# Structural insights into translocation and tailored synthesis of hyaluronan

Ireneusz Górniak ®[1], Zachery Stephens[1], Satchal K. Erramilli[2],
Tomasz Gawda ®[2], Anthony A. Kossiakoff ®[2] & Jochen Zimmer ®[1,3] ✉

Hyaluronan (HA) is an essential component of the vertebrate extracellular matrix. It is a heteropolysaccharide of N-acetylglucosamine (GlcNAc) and glucuronic acid (GlcA) reaching several megadaltons in healthy tissues. HA is synthesized and translocated in a coupled reaction by HA synthase (HAS). Here, structural snapshots of HAS provide insights into HA biosynthesis, from substrate recognition to HA elongation and translocation. We monitor the extension of a GlcNAc primer with GlcA, reveal the coordination of the uridine diphosphate product by a conserved gating loop and capture the opening of a translocation channel to coordinate a translocating HA polymer. Furthermore, we identify channel-lining residues that modulate HA product lengths. Integrating structural and biochemical analyses suggests an avenue for polysaccharide engineering based on finely tuned enzymatic activity and HA coordination.

Hyaluronan (HA) is an extracellular matrix polysaccharide of vertebrate tissues, with essential functions in osmoregulation, cell signaling and joint lubrication[1,2]. HA's physiological functions are influenced by its length and gel-forming properties[3]. Healthy tissues usually contain polymers of several megadaltons in size[2]. Inflammatory processes and osteoarthritis, however, are associated with a decline in HA length to hundreds of kilodaltons[4,5]. Humans express three HAS isoforms, with HAS2 being essential[6]. While HAS1 and HAS2 produce HA polymers of several megadaltons, HAS3-derived products usually range from $10^5$ to $10^6$ Da in size[7].

HA consists of alternating N-acetylglucosamine (GlcNAc) and glucuronic acid (GlcA) units[8] and is synthesized and translocated across the plasma membrane by HA synthase (HAS), a membrane-embedded processive glycosyltransferase (GT) (Fig. 1a,b)[9–11]. HAS integrates several functions: (1) binding its substrates (uridine diphosphate (UDP)-activated GlcA and GlcNAc); (2) catalyzing glycosyl transfer; and (3) translocating the nascent polymer across the plasma membrane. How these functions are integrated for HA biosynthesis is poorly understood.

While HA is ubiquitously expressed in vertebrates, the *Paramecium bursaria Chlorella* virus (PBCV) encodes an HAS similar to the vertebrate homologs[12]. Structural and functional analyses of *Chlorella* virus (Cv)HAS provided important insights into the enzyme's architecture and how it initiates HA biosynthesis[13]. However, because the enzyme produces HA polymers substantially shorter than vertebrate HAS, in addition to lacking a nascent HA chain in structural analyses, little was learned about how HAS coordinates the nascent HA polymer, translocates it between elongation steps and regulates its size.

Biochemical and structural analyses of *Xenopus laevis* HAS isoform 1 (XlHAS1, formerly DG42)[14–16] address these important aspects of HA biosynthesis. In vitro, XlHAS1 produces polysaccharides similar in size to natively expressed HA. Cryogenic electron microscopy (cryo-EM) analyses of the enzyme bound to a nascent HA chain reveal how it creates an HA translocation channel and coordinates the polysaccharide during translocation. Complementary cryo-EM structures of CvHAS bound to the substrate UDP-GlcA and reaction products provide insights into alternating substrate polymerization and suggest a model for HA translocation. Furthermore, site-directed mutagenesis of channel-lining residues identifies important interactions modulating HA length.

## Results

### XlHAS1 synthesizes HA polymers in a megadalton size range
XlHAS1 was expressed in *Spodoptera frugiperda* (Sf9) cells and purified using immobilized metal affinity chromatography (IMAC) and size-exclusion chromatography (SEC) in glyco-diosgenin (GDN)

[1]Department of Molecular Physiology and Biological Physics, University of Virginia, Charlottesville, VA, USA. [2]Department of Biochemistry and Molecular Biology, University of Chicago, Chicago, IL, USA. [3]Howard Hughes Medical Institute, Chevy Chase, MD, USA. ✉e-mail: jz3x@virginia.edu

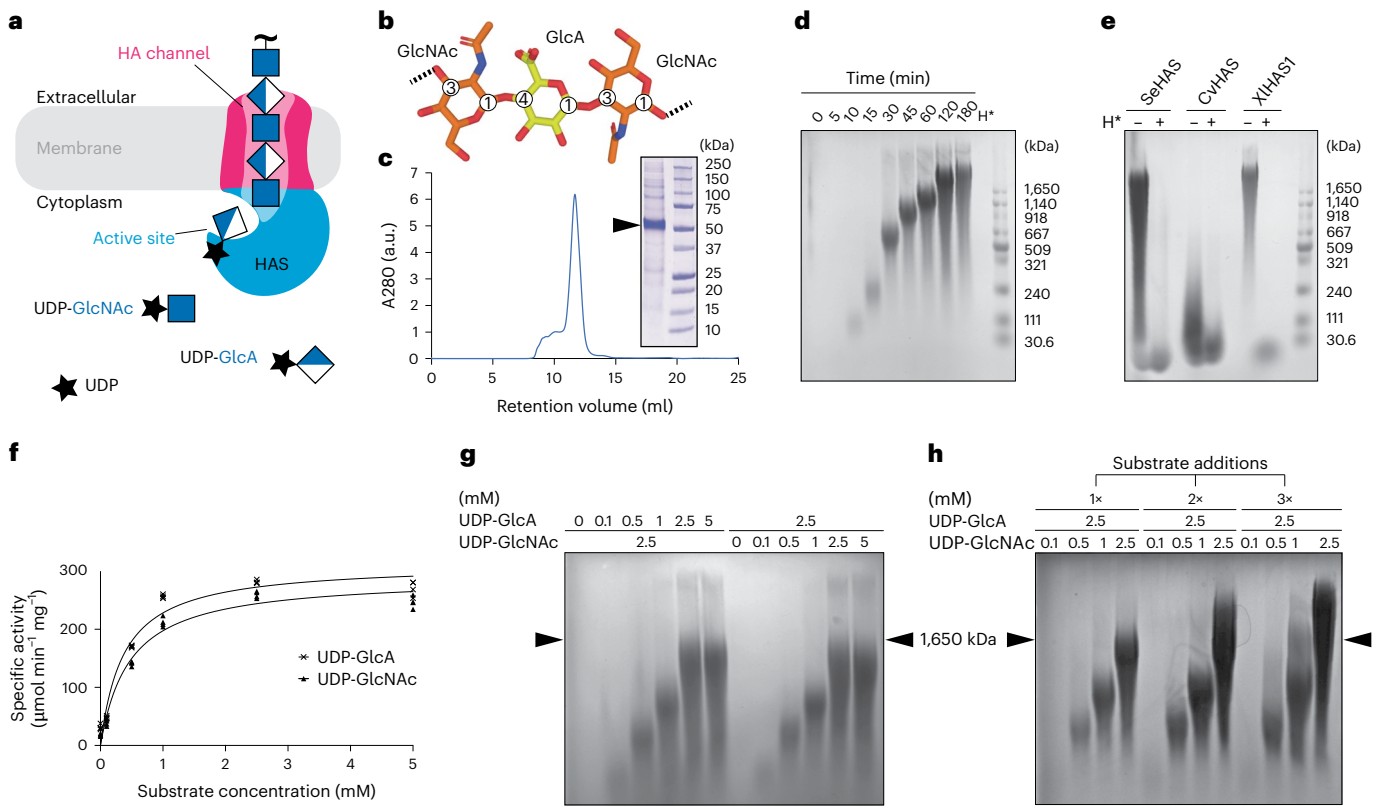

**Fig. 1 | In vitro HA biosynthesis. a**, HAS couples HA synthesis with membrane translocation. **b**, HA trisaccharide structure. **c**, S200 size-exclusion chromatogram of purified XlHAS1. Inset, Coomassie-stained SDS–PAGE. **d**, Time course of in vitro HA biosynthesis monitored by agarose gel electrophoresis and Stains-all staining. H*, hyaluronidase digestion. The molecular weight marker represents HA size standards. **e**, Comparison of HA synthesized in vitro by SeHAS, CvHAS and XlHAS1. All proteins were assayed after purification in the GDN detergent. **f**, Catalytic activity depending on substrate concentration. One

substrate concentration was at 2.5 mM while the other was varied as indicated. Activity was measured by quantifying the released UDP in an enzyme-coupled reaction. **g**, The same as in **f** but by monitoring HA formation. **h**, Increasing HA molecular weight in the absence of product inhibition. A standard synthesis reaction was supplemented three times with 2.5 mM UDP-GlcA and the indicated amounts of UDP-GlcNAc while enzymatically converting UDP to UTP. Arrowheads in **g** and **h** indicate the 1,650-kDa HA marker for orientation. For **c–h**, experiments were performed at least three times with similar results.

detergent (Methods). On the basis of SEC and cryo-EM analyses discussed below, the catalytically active enzyme is monomeric (Fig. 1c).

HAS transfers the UDP-activated glycosyl unit (the donor sugar) to the nonreducing end of the nascent HA polymer (the acceptor sugar). This reaction generates UDP and elongated HA as reaction products (Fig. 1a). Catalytic activity can be assessed by monitoring the release of UDP in an enzyme-coupled reaction, as previously described[10,13], or quantifying HA either radiometrically or by dye staining following electrophoresis.

XlHAS1 exhibits robust catalytic activity in the presence of magnesium cations, whereas its activity is reduced by ~30% when manganese is instead used as the cofactor (Extended Data Fig. 1a,b). The obtained polysaccharide is readily degraded by hyaluronidase, demonstrating that it is indeed authentic HA. Agarose gel electrophoresis of the in vitro reaction product followed by Stains-all staining reveals increasing polymer lengths over a ~120-min synthesis reaction before stalling (Fig. 1d). Notably, at completion, XlHAS1 synthesizes HA polymers of a consistent length that migrate above a 1.6-MDa HA marker. This material is comparable to HA in vitro synthesized by *Streptococcus equisimilis* (Se) HAS. In contrast, CvHAS produces polydisperse low-molecular-weight HA of ~30–200 kDa (Fig. 1e).

Kinetic analyses were performed by titrating one substrate at a saturating concentration of the other and real-time quantification of the released UDP product (Fig. 1f and Extended Data Fig. 1c). The results reveal Michaelis–Menten constants for UDP-GlcNAc and UDP-GlcA of about 470 and 370 μM, respectively, and an overall catalytic rate

of approximately 30 substrate turnovers per minute. Monitoring the length distribution of the synthesized HA under these substrate conditions (one limiting, with the other in excess) shows increasing HA lengths up to a substrate concentration of 2.5 mM (Fig. 1g). On the basis of the observed HA product lengths (~1.6 MDa produced within 90 min; Fig. 1d), the electrophoretic analysis suggests a catalytic rate on the same order of magnitude as determined by UDP quantification, yet about twofold higher for unknown reasons.

In vivo, HA biosynthesis could stall because of substrate depletion, competitive UDP inhibition and/or HA accumulation. To analyze the impact of sustained high substrate concentrations during HA biosynthesis, an in vitro synthesis reaction was supplemented with both substrates 2, 4 and 6 h after initiation, while converting the accumulating UDP to noninhibitory uridine triphosphate (UTP) (Fig. 1h and Extended Data Fig. 1d–f). These conditions further extend the synthesized HA polymers to products substantially exceeding the 1.6-MDa marker. Extension is only observed when both substrates are supplied in excess (2.5 mM). If one substrate (UDP-GlcNAc) remains at a limiting concentration (0.1–1 mM), no increase in HA size is observed, likely because of premature HA release followed by reinitiation.

### Architecture of XlHAS1 in a resting state
We selected a noninhibitory antibody Fab fragment to facilitate cryo-EM structural analyses of XlHAS1 (Fig. 2a,b and Extended Data Fig. 2). The purified XlHAS1–Fab complex was either used directly

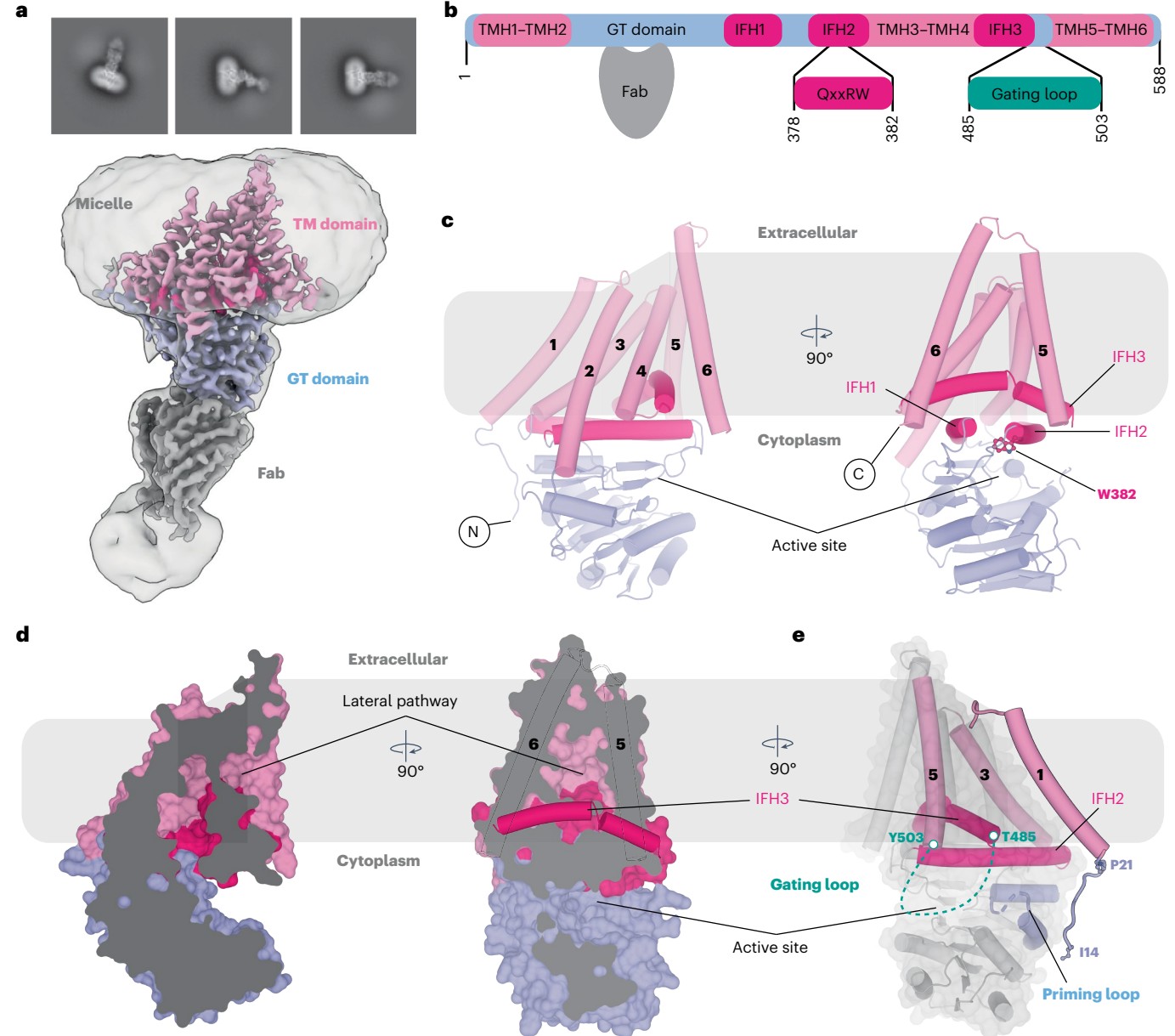

**Fig. 2 | Structure of XlHAS1. a**, Representative 2D class averages and electron potential map of XlHAS1. The map was contoured at $\sigma = 9$ r.m.s.d. (root-mean-square deviation) and is overlaid with an ab initio volume to visualize the detergent micelle. **b**, Domain organization of XlHAS1 and Fab-binding site. **c**, Architecture of XlHAS1. The catalytic domain, amphipathic IFHs and transmembrane regions are colored blue, light pink and pink, respectively. **d**, Surface representation of XlHAS1 indicating a curved channel with lateral pathway. **e**, Cartoon representation of XlHAS1 highlighting the positions of the N-terminal extension and the unresolved gating loop above the active site (dashed line).

for cryo-EM grid preparation or incubated with substrates to obtain snapshots of biosynthetic intermediates (Extended Data Figs. 2–4 and Table 1).

XlHAS1's cytosolic GT domain interacts with two N-terminal and four C-terminal transmembrane helices (TMHs) and three cytosolic interface helices (IFH1–IFH3) (Fig. 2b–e). The IFHs surround the entrance to a curved channel that ends with a lateral portal halfway across the membrane, formed by IFH3, TMH5 and TMH6 (Fig. 2d). The function of this pathway is unknown as it is not used for HA translocation, except for its cytosolic entrance, as discussed below. IFH2 contains the conserved QxxRW motif (Extended Data Fig. 5) of which W382 is pivotal for positioning the acceptor sugar right at the entrance to the transmembrane (TM) channel.

The XlHAS1 architecture is consistent with an AlphaFold (AF)-predicted structure of human HAS1 and HAS2 and the previously determined CvHAS structure (Extended Data Fig. 6a,b)[13], with the exception that TMH1 is clearly resolved in the XlHAS1 maps (Fig. 2c,e and Extended Data Fig. 2c). This curved helix rests against the cytosolic connection of IFH2 and TMH3 (Fig. 2e).

Vertebrate HASs contain a conserved WGTSGRR/K motif in a cytosolic loop connecting IFH3 with TMH5. A similar motif in a loop close to the active site is found in chitin and cellulose synthases[13,17–20] (Extended Data Fig. 4d–f). Although the loop is unresolved in the apo XlHAS1 structure, its flanking residues T485 and Y503 at the C terminus of IFH3 and beginning of TMH5, respectively, position it right above the catalytic pocket (Fig. 2e). In analogy to cellulose

**Table 1 | Cryo-EM data collection, refinement and validation statistics**

| | XlHAS1 Apo (EMD-40591) (PDB 8SMM) | XlHAS1 HA-bound (EMD-40594) (PDB 8SMN) | XlHAS1 UDP-bound (EMD-40598) (PDB 8SMP) | CvHAS primed UDP-GlcA-bound (EMD-40623) (PDB 8SND) | CvHAS HA2-bound (EMD-40622) (PDB 8SNC) | CvHAS HA2+UDP-bound (EMD-40624) (PDB 8SNE) |
|---|---|---|---|---|---|---|
| **Data collection and processing** | | | | | | |
| Magnification | 81,000 | 81,000 | 81,000 | 81,000 | 81,000 | 81,000 |
| Voltage (kV) | 300 | 300 | 300 | 300 | 300 | 300 |
| Electron exposure (e⁻ per Å²) | 50 | 50 | 50 | 50 | 50 | 50 |
| Defocus range (μm) | −2.0 to −1.0 | −2.0 to −1.0 | −2.0 to −1.0 | −2.0 to −1.0 | −2.0 to −1.0 | −2.0 to −1.0 |
| Pixel size (Å) | 1.08 | 1.08 | 1.08 | 1.08 | 1.08 | 1.08 |
| Symmetry imposed | *C1* | *C1* | *C1* | *C1* | *C1* | *C1* |
| Initial particle images (no.) | 6,543,217 | 22,061,285 | 26,290,833 | 7,978,429 | 3,663,447 | 12,554,677 |
| Final particle images (no.) | 337,022 | 169,894 | 136,942 | 176,543 | 56,954 | 220,556 |
| Map resolution (Å) | 3.2 | 3.0 | 3.2 | 3.2 | 3.3 | 3.0 |
| FSC threshold | 0.143 | 0.143 | 0.143 | 0.143 | 0.143 | 0.143 |
| Map resolution range (Å) | 2.7–6.0 | 2.6–3.5 | 2.8–5.0 | 2.4–3.5 | 2.9–4.5 | 2.3–3.4 |
| **Refinement** | | | | | | |
| Initial model used (PDB code) | AF2 B1WB39 | 8SMM | 8SMM | 7SP6 | 7SP6 | 7SP6 |
| Model resolution (Å) | 3.2 | 3.0 | 3.2 | 3.2 | 3.3 | 3.0 |
| FSC threshold | 0.143 | 0.143 | 0.143 | 0.143 | 0.143 | 0.143 |
| Map sharpening *B* factor (Å²) | 95.4 | 102.0 | 87.0 | 133.0 | 117.7 | 108.6 |
| **Model composition** | | | | | | |
| Nonhydrogen atoms | 5,930 | 6,138 | 6,097 | 6,090 | 5,928 | 6,025 |
| Protein residues | 738 | 749 | 757 | 742 | 728 | 735 |
| Ligands | 0 | 9 | 2 | 6 | 6 | 6 |
| **B factors (Å²)** | | | | | | |
| Protein | 75.6 | 35.98 | 73.11 | 68.63 | 58.44 | 52.18 |
| Ligand | - | 50.07 | 60.99 | 82.53 | 54.72 | 55.03 |
| **R.m.s.d.** | | | | | | |
| Bond lengths (Å) | 0.004 | 0.002 | 0.002 | 0.005 | 0.004 | 0.008 |
| Bond angles (°) | 0.660 | 0.519 | 0.526 | 0.985 | 0.859 | 1.054 |
| **Validation** | | | | | | |
| MolProbity score | 1.63 | 1.48 | 1.64 | 1.20 | 1.58 | 1.54 |
| Clashscore | 6.97 | 4.61 | 6.29 | 4.23 | 5.36 | 4.92 |
| Poor rotamers (%) | 0.31 | 0.15 | 0.3 | 0.0 | 0.32 | 0.16 |
| **Ramachandran plot** | | | | | | |
| Favored (%) | 96.28 | 96.35 | 95.72 | 98.22 | 95.81 | 95.86 |
| Allowed (%) | 3.72 | 3.65 | 4.15 | 1.64 | 4.19 | 4.00 |
| Disallowed (%) | 0.00 | 0.00 | 0.13 | 0.14 | 0.00 | 0.14 |

FSC, Fourier shell correlation.

synthase and functional analyses described below, we refer to this loop as XlHAS1's 'gating loop'.

**Nascent HA coordination inside an electropositive channel**

To stabilize an HA-associated translocation intermediate, we performed an in vitro HA synthesis reaction in the presence of HA lyase to trim polymers emerging from the HAS TM channel. Cryo-EM maps of HA-bound XlHAS1 reveal continuous nonproteinaceous density running from the catalytic pocket to its extracellular surface through an electropositive channel (Fig. 3a and Extended Data Fig. 3c). The density accommodates nine glycosyl units, denoted 1–9 starting at the nonreducing end at the acceptor position. The assignment of the HA

register is based on three observations. Firstly, the shape of the nonreducing end terminal glycosyl unit resembles a GlcNAc moiety and is consistent with maps of GlcNAc-primed CvHAS (Extended Data Fig. 3c). Secondly, following this register, the densities at the polymer's third and fifth glycosyl positions are also consistent with GlcNAc moieties, as expected for an alternating GlcNAc–GlcA repeat unit. This register is also consistent with previous analyses of CvHAS demonstrating stable coordination of GlcNAc but not GlcA at the acceptor position[13]. Thirdly, the carboxylate substituents of the GlcA units are expected to be weak or even absent in the electron potential maps (Extended Data Fig. 3c). Despite confidence in the register of the modeled HA chain, the exact orientation of the glycosyl units is less certain. Disaccharide repeat

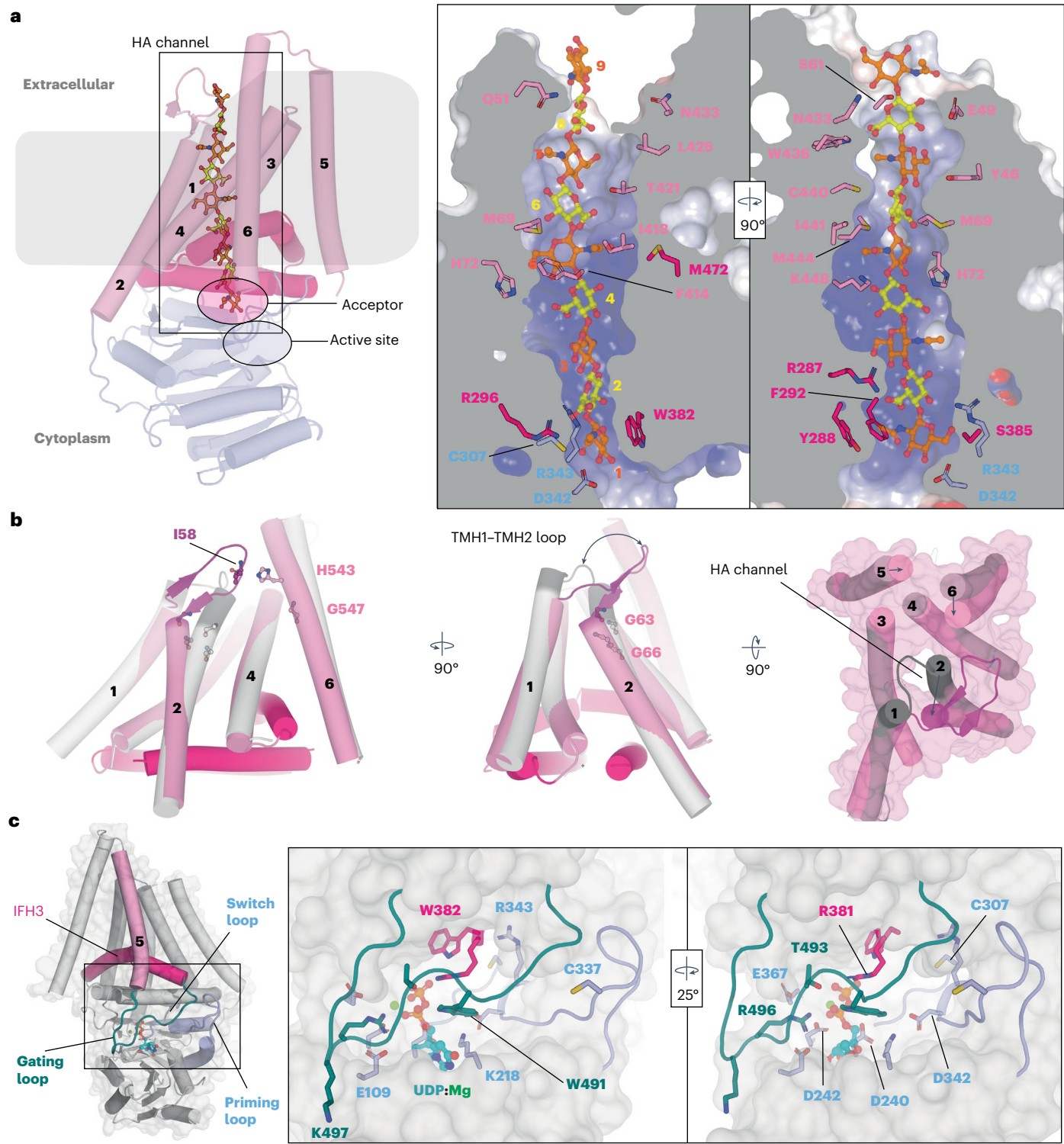

**Fig. 3 | An HA–HAS translocation intermediate. a**, Left, XlHAS1 structure with a nascent HA nonasaccharide. Right, stick representations of residues coordinating HA inside XlHAS1 TM channel colored by the electrostatic surface potential calculated in PyMOL using the APBS plugin[36] (red to blue, −10 to 10 kT). The nascent HA polymer is shown in ball-and-stick representation with carbons colored yellow and orange for GlcA and GlcNAc units, respectively. **b**, Gating transitions upon HA formation. Shown is a superimposition of the TM regions only in the presence (pink) and absence (white) of a translocating HA polymer. **c**, Gating loop insertion into the catalytic pocket. Conserved residues contacting UDP are shown as sticks. UDP is shown in ball-and-stick representation colored cyan for carbon atoms.

units of the HA polymer likely enter the translocation channel in two orientations with their acetamido and carboxylate groups pointing roughly in opposite directions (Supplementary Discussion and Supplementary Fig. 1). Thus, in addition to the modeled conformation, we

cannot exclude contributions from an alternatively oriented polymer to the observed cryo-EM map.

The first GlcNAc (GlcNAc-1) sits inside a collar of invariant residues, including Y288, F292, C307, W382 and S385, as well as D342 and R343 of

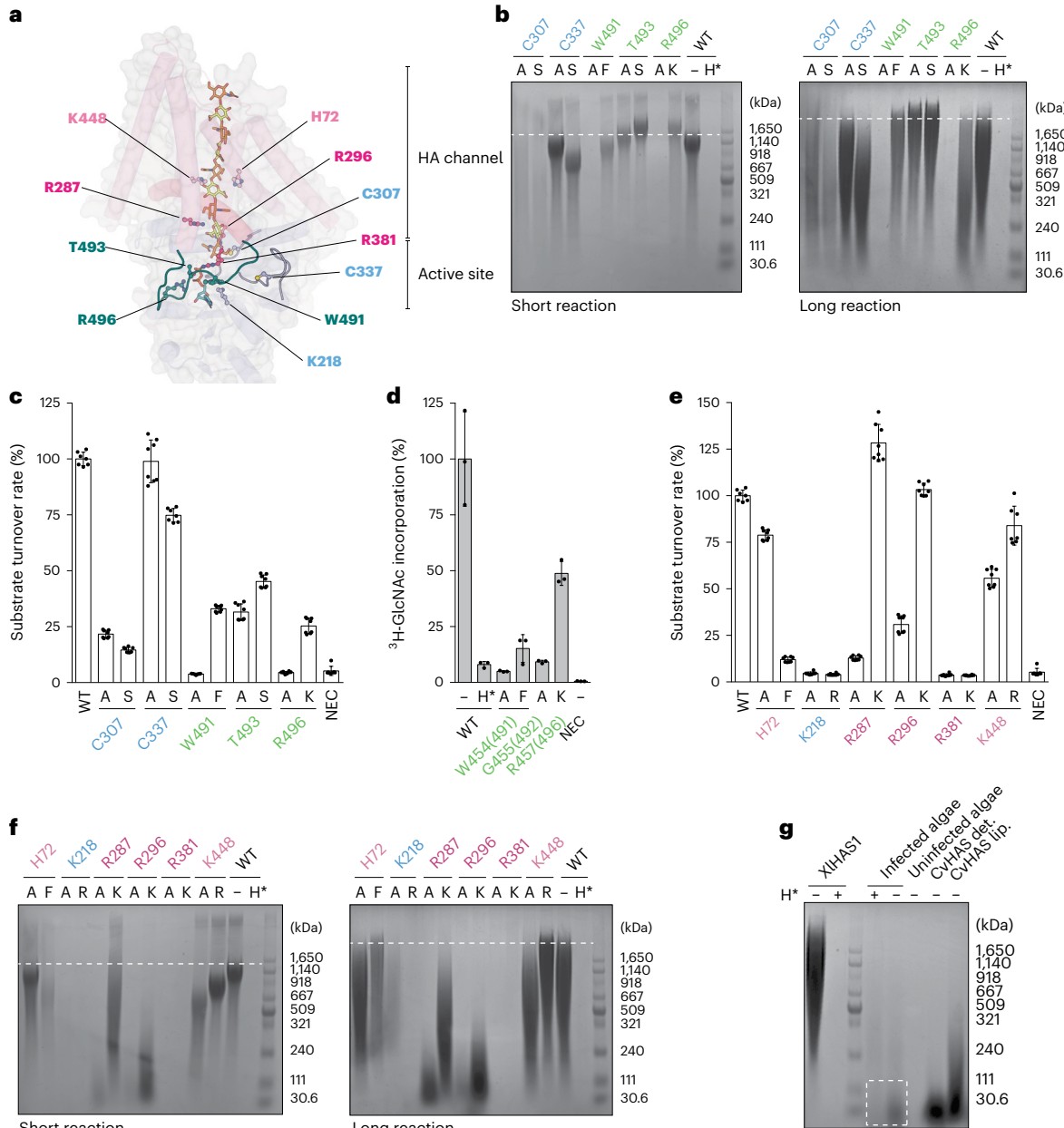

**Fig. 4 | Modulation of HA length. a**, XlHAS1 composite model showing mutagenized residues forming the active site (colored blue and teal for residues of the GT domain and gating loop, respectively) or the HA channel (colored light pink and pink for residues of TMHs and IFHs, respectively). **b**, Mutagenesis of active-site-lining residues. In vitro HA biosynthesis by the indicated XlHAS1 mutants for 1 h or 8 h (short or long reaction, respectively). Agarose gels were stained with Stains-all. The dashed line roughly indicates the WT's highest-molecular-weight product. **c**, Catalytic rates of the mutants shown in **b**. Rates, normalized to WT, report substrate turnovers and were determined by quantifying UDP release in real time during HA synthesis (*n* = 8). **d**, Catalytic activity of gating loop mutants of CvHAS. Activity was determined

by quantifying ³H-labeled HA by scintillation counting and is expressed relative to WT[10]. Numbers in parentheses indicate residue numbers in XlHAS1 (*n* = 3). **e**, Mutagenesis of charged channel and active-site-lining residues. In vitro HA biosynthesis by the indicated mutants, similar to **c** (*n* = 8). NEC, no-enzyme control. Error bars in **c**–**e** represent the s.d. from the mean. **f**, Similar to **b** but for charged residues. **g**, Comparison of HA products obtained from NC64A *Chlorella* algae infected with PBCV-1 virus and in vitro synthesized by purified XlHAS1 and CvHAS in LMNG + CHS micelle (det.) and proteoliposome (lip.). The boxed region indicates the hyaluronidase-sensitive isolated sample. The experiment was repeated three times with similar results.

the conserved GDDR motif (Fig. 3a and Extended Data Fig. 5). Following the acceptor site, the conserved R296 and R287 of IFH1 are in close proximity to GlcA-2 and likely stabilize GlcA units inside the channel. Past this point, the channel dimensions widen and HA's interactions with side chains are less extensive.

While HA's first three glycosyl units enter the channel in a coplanar conformation, the following GlcAc-4–GlcNAc-5 disaccharide is roughly 90° out of plane (Fig. 3a and Extended Data Fig. 3c). The rotation is

evident from the planes of the resolved glycopyranose rings. It occurs at a central widening of the TM channel that can accommodate spontaneous structural rearrangements of the polysaccharide, as observed in solution[21]. At this widening, H72 and K448 contribute to the channel's electropositive character (Fig. 3a).

Similarly, the next disaccharide unit (GlcA-6 and GlcNAc-7) is rotated by about 45° relative to the preceding glycosyl units and the following disaccharide (GlcA-8 and GlcNAc-9) exhibits a similar rotation

relative to the preceding pair (Extended Data Fig. 3c). Past GlcNAc-9, insufficient EM map quality prevents further interpretations.

Of note, about halfway across the membrane, the HA polymer is encircled by a methionine-rich ring of hydrophobic residues, including M69, F414, I418, I441, M444 and M472 (Fig. 3a). Past this hydrophobic ring, the nascent HA chain is surrounded by moderately conserved hydrophilic and hydrophobic residues, including Y46, E49, Q51, S61, T421, L425, N433, W436 and C440 (Fig. 3a). This channel segment is strikingly devoid of positively charged residues, unlike the preceding section.

### Bending of TMHs opens the HA translocation channel
HA translocation requires conformational changes of XlHAS1 to create a continuous TM channel. Compared to the resting conformation described above, the N-terminal half of TMH2 near the extracellular water–lipid interface bends away from TMH4 (Fig. 3b). Bending occurs around a conserved GLYG motif (residues 63 to 66) that places two glycine residues at the interface with TMH4. Furthermore, the extracellular helical turns of TMH1 and TMH2 unwind to form a short β-hairpin with residues of the TMH1–TMH2 loop (Fig. 3b). Although only the loop's backbone is resolved in XlHAS1's apo conformation (Extended Data Fig. 2c), it is evident that the loop lids the extracellular channel exit in the absence of HA. Upon channel opening, the loop flips toward the membrane and rotates by approximately 90° to run roughly parallel to the membrane surface. Additionally, the N-terminal half of TMH6 bends by approximately 10° toward TMH2, around the conserved G547 (Fig. 3b). In the new position, H543 is in hydrogen-bonding distance to the backbone nitrogen of I58. Compared to the open conformation of CvHAS observed in the presence of a GlcNAc primer[13], the unwinding of the N-terminal segment of TMH2 in XlHAS1 creates a larger extracellular channel portal. Furthermore, the bending of TMH6 is not observed in the HA-free open channel conformation of CvHAS (Extended Data Fig. 6d).

### The HAS gating loop inserts into the catalytic pocket
Attempts to stabilize an HA translocation intermediate also resulted in an HA-free but UDP-bound conformation (Fig. 3c and Extended Data Fig. 4). In this structure, UDP is coordinated by conserved motifs of the GT domain, as previously described for CvHAS and other processive and nonprocessive GTs of the GT-A fold[13,17,18,22–24]. Importantly, the map resolves XlHAS1's gating loop, connecting IFH3 with TMH5 (residues 485 to 503) and containing the conserved WGTSGRK/R sequence (residues 491 to 497) (Extended Data Figs. 4 and 5). The loop inserts into the catalytic pocket and interacts with the nucleotide (Fig. 3c). In this conformation, W491 forms a cation–π interaction with R381 of the QxxRW motif in IFH2, which in turn forms a salt bridge with the nucleotide's diphosphate group. The indole ring of W491 runs approximately perpendicular to the uracil moiety, placing its Nε within hydrogen-bonding distance to UDP's α-phosphate. This phosphate group is also contacted by the side chain of the following T493. R496, the penultimate residue of the WGTSGRK/R motif inserts into a negatively charged pocket formed by E109, D242, E367 and UDP's α-phosphate (Fig. 3c).

### The gating loop is necessary for HA biosynthesis
Site-directed mutagenesis was used to reveal the gating loop's functional importance in two model systems, XlHAS1 and CvHAS (Fig. 4). For XlHAS1, replacing W491 of the **W**GTSGRK/R motif with alanine (W491A) abolishes HA biosynthesis during a 60-min synthesis reaction (denoted 'short reaction'), while its substitution with phenylalanine (W491F) results in a lower-molecular-weight product (Fig. 4a,b). Similarly, quantifying UDP release by these mutants shows abolished and reduced substrate turnover rates, respectively, compared to the wild-type (WT) enzyme (Fig. 4c). Interestingly, over an 8-h synthesis reaction ('long reaction'), the W491F mutant can produce HA polymers similar in size to the WT enzyme. Likewise, replacing the following

T493 (WG**T**SGRK/R) with alanine or serine reduces the catalytic rate to about 20–25% of that of the WT enzyme (Fig. 4c). Strikingly, the T493A and T493S mutants generate HA exceeding the size of the WT product even during a 'short' synthesis reaction (Fig. 4b). The size differences are more pronounced on a lower percentage agarose gel (Extended Data Fig. 6e). Furthermore, replacing R496 (WGTSG**R**K/R) with alanine abolishes catalytic activity, while an R496K mutant produces polymers of slightly increased length and greater polydispersity in short and long synthesis reactions, respectively (Fig. 4b).

Similar results were obtained for CvHAS by quantifying HA using scintillation counting (Fig. 4d). Here, the gating loop contains a WGTRG sequence. Replacing the motif's tryptophan residue (W454) with alanine or phenylalanine is incompatible with function and so is the substitution of the following G455 with alanine (WG**T**RG). Replacing the arginine residue (R457) with lysine reduces CvHAS activity by ~60% relative to WT.

### HA coordination modulates polymer length
Processive HA biosynthesis requires sustained HAS–HA interactions between elongation steps. To test how HA coordination affects the HA length distribution, we altered active-site-lining and channel-lining residues and monitored HA size and UDP release.

XlHAS1's active site contains two conserved cysteines. C307 belongs to the 'switch loop' at the back of the active site, adjacent to the GlcNAc acceptor (Figs. 3c and 4a). Replacing C307 with alanine or serine leads to the production of low amounts of polydisperse HA (Fig. 4b). The second cysteine, C337, is part of a flexible 'priming loop' located to one side of the catalytic pocket (Figs. 3c and 4a). This residue can be replaced with alanine or serine, resulting in HA products similar to WT, with a slight length reduction for the C337S mutant (Fig. 4b).

Also in the active site, replacing the conserved K218 with alanine or arginine abolishes substrate turnover and HA synthesis (Fig. 4e,f). Similarly, R381 of the QxxRW motif, located in IFH2 right above the catalytic pocket, cannot be replaced with alanine or lysine. This residue interacts with the nucleotide's pyrophosphate group in UDP-bound and substrate-bound states, as discussed below[25].

Inside the TM channel, the polymer's GlcA-2 is near R287 and R296 of IFH1 (Figs. 3a and 4a). Substituting these residues with alanine reduces substrate turnover rates to about 15% and 30%, respectively, compared to WT, resulting in the generation of short HA oligosaccharide species (Fig. 4e,f). Replacing the residues with lysine, however, results in HAS mutants with similar or even increased substrate turnover rates (compared to WT) that produce polydisperse (R287K) and low-molecular-weight (R296K) HA products in short and long synthesis reactions (Fig. 4e,f).

Farther inside the channel, H72 and K448 are adjacent to GlcA-4. Compared to the WT enzyme, the H72A, K448A and K448R mutants display ~50–75% and the H72F substitution displays ~10% catalytic activity (Fig. 4e). In short synthesis reactions, all mutants produce HA polymers of reduced length, with the H72A product being closest to WT (Fig. 4f). In a long synthesis reaction, however, the H72F and K448R variants produce polymers equivalent to or exceeding the HA length obtained from the WT enzyme (Fig. 4f and Extended Data Fig. 6d). The observed differences in electrophoretic mobility suggest an HA size difference of up to 0.5 MDa for some mutants, including T493S and K448R (Fig. 4b,f and Extended Data Fig. 6e).

Attempts to modulate the product size distribution of CvHAS by stabilizing the enzyme's TM architecture were unsuccessful, resulting in similar product profiles. Isolating natively produced HA from infected *Chlorella* algae confirmed the production of low-molecular-weight HA in vivo (Fig. 4g and Extended Data Fig. 6f,g). Comparing CvHAS polymers produced in a detergent micelle or liposome environment (using *Escherichia coli* total lipid extract) identifies a moderate increase in HA length when produced from proteoliposomes or in a GDN micelle, compared to the native product. It is likely that the different membrane mimetics have different stabilizing or destabilizing effects on the

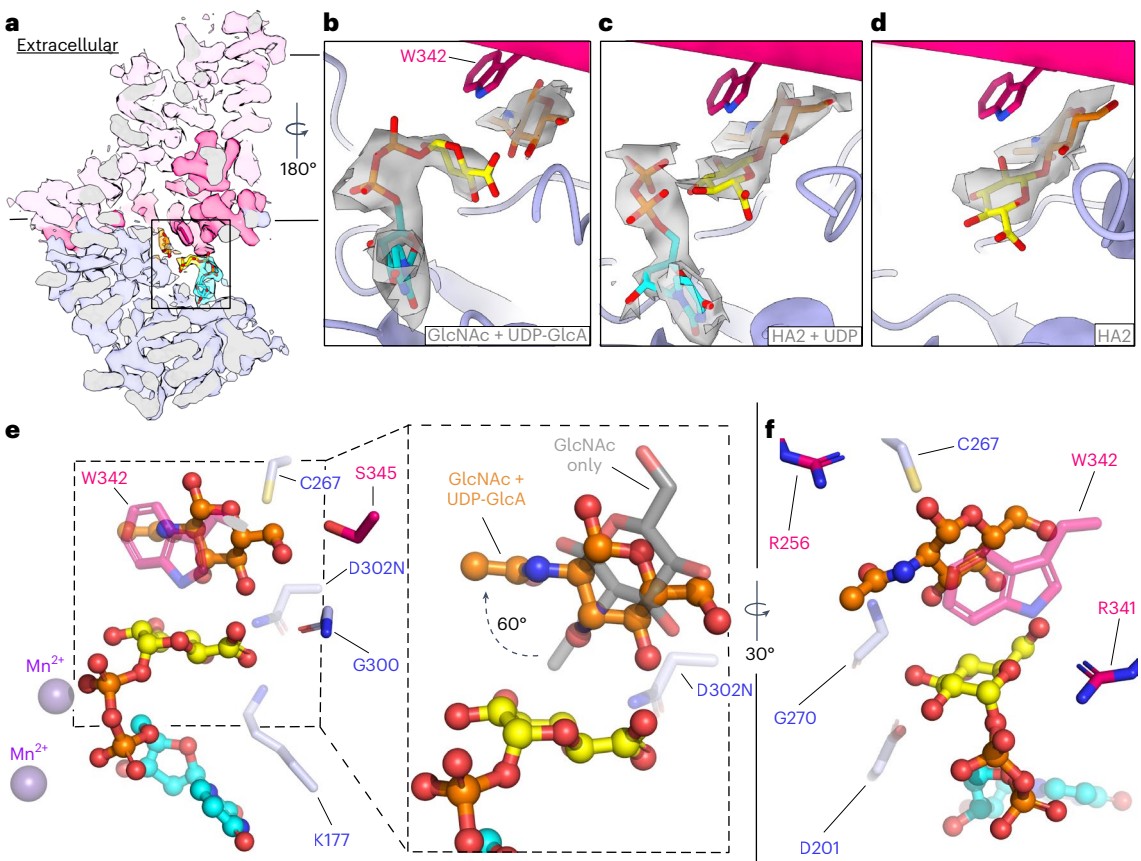

**Fig. 5 | Substrate binding and primer extension by CvHAS. a,** Cross-sectional view of the cryo-EM density map for primed CvHAS bound to UDP-GlcA. The priming GlcNAc is colored orange and the UDP-GlcA substrate is colored yellow and cyan for the carbon atoms of the donor sugar and uracil moiety, respectively, as are the densities. **b–d,** Extension of the GlcNAc primer by GlcA. Densities for GlcNAc and UDP-GlcA, $\sigma$ = 4.5 r.m.s.d. (**b**), HA disaccharide (HA2) and UDP, $\sigma$ = 4.0 r.m.s.d. (**c**) and HA2 only, $\sigma$ = 4.5 r.m.s.d. (**d**), shown as transparent gray surfaces. **e,f,** CvHAS active-site interactions with GlcNAc and UDP-GlcA. GlcNAc and UDP-GlcA are shown as ball-and-stick models, while side chains are shown as sticks only. Inset, the GlcNAc primer position in the absence of a UDP-GlcA substrate (PDB 7SP9) as gray sticks.

CvHAS transmembrane domain and, thus, HA coordination. The longer CvHAS-produced polymers are comparable to products obtained from the XlHAS1 R287A, R296A and R296K mutants (Fig. 4f,g and Extended Data Fig. 6g), as discussed below.

### Substrate binding repositions the GlcNAc primer

Upon priming with a GlcNAc monosaccharide[13], HAS attaches GlcA to the C3 hydroxyl group of the primer to form the GlcA–GlcNAc disaccharide repeat unit. To gain insights into this step, we took advantage of the high-quality cryo-EM maps routinely obtained for CvHAS, facilitated by camelid nanobodies[13]. The catalytically inactive CvHAS D302N mutant[13,26] in complex with two nanobodies was incubated with GlcNAc and Mn$^{2+}$:UDP-GlcA before grid preparation (Extended Data Fig. 7 and Table 1). In the resulting cryo-EM map, the GlcNAc primer is well resolved and occupies the same acceptor position as previously reported, stacking against W342 (Fig. 5a,b and Extended Data Fig. 7)[13]. In the presence of the UDP-GlcA substrate, however, the primer rotates by approximately 60°, placing its acetamido group away from the active site (Fig. 5e,f). This rotation positions the primer's C3 hydroxyl group closer to the putative base catalyst D302 (replaced with asparagine in this construct). In this position, the acetamido group's carbonyl oxygen is in proximity to the backbone amide nitrogen of G270. The sugar's ring oxygen is adjacent to the sulfhydryl group of the conserved C267, its C1 hydroxyl may form a water-mediated interaction with R256 and the C6 hydroxyl group is in proximity to S345. All acceptor-surrounding residues are conserved.

CvHAS binds both of its substrates in similar binding poses, with UDP being coordinated by two manganese ions, as previously described[13] (Fig. 5e,f). The donor GlcA sugar sits roughly underneath W342 of the QxxRW motif. Its C6 carboxylate is incompletely resolved, as often observed in electron potential maps. In this binding pose, however, the carboxylate occupies a pocket formed by the C-terminal end of the priming loop (residues 298 to 300), the following helix that begins with the invariant GDDR motif (residues 300–303) and the priming GlcNAc sugar (Fig. 5e). Furthermore, R341 of the QxxRW motif is in close proximity to the donor's carboxylate (Fig. 5f). Additional residues surrounding the donor sugar include D201, K177 and the introduced N302 (D302N). The distance between the acceptor's C3 hydroxyl group and the donor's C1 carbon atom is long (about 5.2 Å), indicating that additional conformational changes are necessary for glycosyl transfer.

### Insights into stepwise HA elongation

Binding of UDP-GlcA to GlcNAc-primed HAS generates an HA disaccharide with GlcA at the nonreducing end. This step was visualized in two experiments. First, exposure of WT CvHAS to both of its substrates resulted in a UDP–HA disaccharide-bound conformation, interpreted as a state directly after glycosyl transfer and before UDP release (Fig. 5c, Extended Data Fig. 8 and Table 1). The density at the acceptor site is consistent with GlcNAc, preceded by weaker density inside the catalytic pocket, assigned to GlcA. Because the UDP and HA disaccharide densities are not clearly separated, we cannot exclude the presence of an

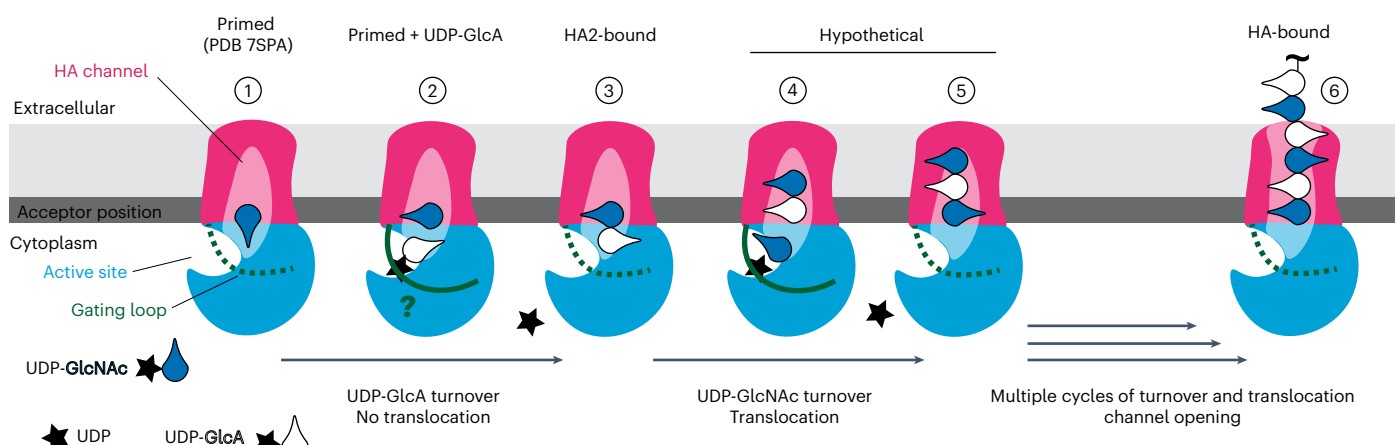

**Fig. 6 | Model of HA synthesis and translocation.** (1) A GlcNAc monosaccharide released from UDP-GlcNAc primes HAS. (2) Coordination of UDP-GlcA at the active site repositions the primer to accommodate the donor glycosyl unit. Transient interactions between the nucleotide and the gating loop likely stabilize the loop (dashed versus solid green line) and contribute to catalytic activity. (3) Glycosyl transfer and UDP release generate a disaccharide-bound state with GlcA extending into the catalytic pocket. (4) UDP-GlcNAc binding correlates with disaccharide translocation to position GlcA at the acceptor site. (5) Glycosyl transfer and UDP release produce a trisaccharide that translocates into the channel after rotation of the nonreducing end terminal GlcNAc unit. (6) After multiple cycles of substrate turnover, the nascent HA chain induces channel opening (see Supplementary Discussion for more information).

overlapping minor second state in which CvHAS is bound to UDP-GlcA and a GlcNAc monosaccharide primer (released from UDP-GlcNAc).

This situation was improved in a second experiment. CvHAS was primed with a GlcNAc monosaccharide and then incubated with UDP-GlcA for elongation with GlcA. The obtained cryo-EM map resolves a GlcNAc at the acceptor position that is elongated by a single sugar unit extending into the catalytic pocket (Fig. 5d, Extended Data Fig. 9 and Table 1). No UDP density is observed in the active site; thus, this pose likely represents a state after glycosyl transfer and UDP release. Although clearly attached to the GlcNAc moiety, the extending GlcA sugar is flexible because of sparse coordination inside the catalytic pocket. We note that the carboxylate group of GlcA, either when extending the priming GlcNAc unit or as part of the donor sugar, likely occupies the same pocket of the catalytic site.

## Discussion

HA, cellulose and chitin synthases are multitasking enzymes that synthesize high-molecular-weight extracellular polysaccharides. Mechanisms by which the enzymes control product lengths and, thus, the polymers' physical properties are largely unresolved. The narrow size distribution of the in vitro XlHAS1-synthesized HA underscores high processivity of biosynthesis. XlHAS1's product size distribution broadens notably during prolonged in vitro synthesis reactions, perhaps because of substrate depletion and/or product inhibition.

Processive HA biosynthesis requires a stable HA–HAS association. Accordingly, modulating HA coordination inside the translocation channel profoundly affects the HA length distribution. Two arginine-to-lysine substitutions within XlHAS1's IFH1 (R287K and R296K) abolish length control or lead to early termination of HA biosynthesis. These residues at the entrance to the TM channel are likely critical in stabilizing the nascent chain for elongation. Additionally, replacing K448, about halfway across the translocation channel, with alanine or arginine substantially reduces the enzyme's catalytic rate, leading to shorter yet discrete product lengths. Notably, substitutions reducing XlHAS1's catalytic rate while not affecting its processivity give rise to higher-molecular-weight HA. In vivo, similar effects may be achieved by limiting substrate availability and/or post-translational modifications of the enzyme[27,28].

Collectively, the observed product size distributions synthesized by the ensemble of XlHAS1 mutants reflect the HA size range produced by vertebrate HAS isoenzymes[2,7]. Although the residues analyzed in this study are conserved across HAS isoforms, channel dynamics and, thus, HA coordination may differ between isoforms. Thus, physiological differences in HA size likely arise from variations in substrate availability, HA coordination and/or metabolic states of the expressing cells and tissues, as previously suggested[29].

Among the HASs studied biochemically and structurally so far, the CvHAS TM architecture is likely most flexible. This is evidenced by its unresolved N-terminal TMH1 together with the loop connecting it with TMH2. In XlHAS1, TMH1 is stabilized through interactions of Y46 in TMH1 with T421 in TMH3, which are conserved among vertebrate HASs (Extended Data Fig. 5). The flexibility of the CvHAs N-terminal region creates a large lateral opening between TMH2 and TMH4. Assuming that the flexibility of this region persists during HA biosynthesis, the lateral exit may facilitate the early release of short HA polymers by CvHAS. Modulation of the CvHAS product size in response to different membrane mimetics supports this interpretation. In contrast, the TMH1–TMH2 loop in XlHAS1 closes the lateral exit at the extracellular water–lipid interface, which facilitates processive HA biosynthesis (Extended Data Fig. 6c,d). Recent analyses of HA biosynthesis in subterranean mammals further pinpointed HAS expression levels and hyaluronidase activities as important determinants of HA size[30].

The shape and electropositive character of XlHAS1's TM channel contrast the flat, acidic channel formed by cellulose synthase[22,31]. Because the channel's gate is near the extracellular water–lipid interface, we estimate that channel opening is induced by HA polymers exceeding 3–4 glycosyl units (Fig. 6). In this case, the nascent chain and the channel's central hydrophobic ring likely prevent water flux across the membrane, similar to other polysaccharide secretion systems[32,33]. The observed rotation of the nascent HA chain inside the translocation channel may generate energetically favorable conformations contributing to HA translocation and/or preventing backsliding.

The HAS gating loop resembles the corresponding loops in cellulose and chitin synthases[20]. While the precise function of the loop is unclear, site-directed mutagenesis experiments of HAS and cellulose synthase demonstrate its profound importance for catalytic activity[20]. Controlling its ability to interact with the substrate at the active site could be a regulatory mechanism, similar to bacterial cellulose synthase[19].

Our new viral HAS structures demonstrate that, upon extending a GlcNAc primer with GlcA, the newly added sugar protrudes into the catalytic pocket, without spontaneous translocation into the

transmembrane channel. In this position, the terminal GlcA would overlap with the sugar moiety of the next donor substrate. Therefore, we propose that binding of a new substrate molecule translocates the GlcA-extended nascent chain into the channel to avoid steric clashes (Fig. 6). The new substrate molecule (UDP-GlcNAc) may transiently stabilize GlcA at the acceptor position.

Extending a GlcA acceptor with GlcNAc would generate an HA polymer with GlcA at the acceptor position. Previous molecular dynamics simulations revealed that this register is unstable, leading to spontaneous HA translocation to place the added GlcNAc moiety at the acceptor site (Fig. 6)[13]. Such a register-dependent spontaneous HA translocation is consistent with the experimental observations that GlcNAc-terminated polymers end at the acceptor site and that this binding pose is not changed after elongation with a GlcA unit (Fig. 6).

Considering the narrow entry into the HA translocation channel, HA must enter the channel in a flat, ribbon-like conformation. This is only possible if newly added glycosyl units can rotate around the glycosidic linkage within the active site to adopt a favorable conformation (Fig. 6). A similar rearrangement has been suggested for terminal glucosyl units in cellulose synthase[34,35]. The register-dependent rearrangements would result in different orientations of the HA disaccharide units within the polymer (Supplementary Discussion). Accordingly, our experimental HA density may reflect an average of two HA orientations. The channel dimensions widen sufficiently past the first three glycosyl units to accommodate further structural rearrangements of HA. Taken together, our structural and functional analyses provide a molecular framework for engineering polysaccharide biosynthesis systems for a plethora of biomedical, agricultural and tissue-engineering purposes.

## Online content

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

## Methods

### XlHAS1 expression

The synthetic gene encoding N-terminally 12xHis-tagged XlHAS1 (UniProt P13563) was cloned into the pACEBac-1 vector using BamHI and HindIII restriction sites. XlHAS1 mutants were generated using QuickChange and Q5 site-directed mutagenesis methods[37,38] with mutagenetic oligos (Supplementary Table 1). Cloning was confirmed by restriction analyses and DNA sequencing. Baculoviruses harboring each target gene were prepared as previously described[31]. Briefly, the XlHAS1-containing pACEBac-1 plasmid was transformed into *E. coli* DH10MultiBac cells. Bacmids for the WT and all XlHAS1 mutants were purified from three white colonies and transfected into Sf9 cells at $1 \times 10^6$ cells per ml using the FuGene reagent (Promega). Sf9 cells (Expression Systems, 94-001F, not authenticated) were maintained in ESF921 medium (Expression Systems) at 27 °C with mild shaking. The baculovirus was amplified to generate P2 virus stock, which was used at 1.5% culture volume to infect Sf9 cells at a density of $3 \times 10^6$ cells per ml. Cells grown in 1-L media bottles (0.45 L per bottle) were pelleted by centrifugation (3,000$g$ for 10 min) 48 h after infection and resuspended in 40 ml of buffer A (40 mM Tris-HCl pH 7.8, 150 mM NaCl, 10% glycerol and 5 mM MgCl$_2$) per 0.9 L of cell culture. The harvested cells were flash-frozen in liquid nitrogen and stored at −80 °C for subsequent use.

### CvHAS expression

Expression of CvHAS was performed as previously described[13]. Briefly, electrocompetent *E. coli* C43 cells were transformed with a pET28a-CvHAS expression vector. Luria–Bertani (LB) medium supplemented with 50 μg ml$^{-1}$ kanamycin was inoculated with the transformed C43 cells and grown overnight in an orbital shaker at 37 °C, 220 r.p.m. The overnight culture was used to inoculate 4 L of terrific broth supplemented with 1xM salts (25 mM Na$_2$HPO$_4$, 25 mM KH$_2$PO$_4$, 50 mM NH$_4$Cl and 5 mM Na$_2$SO$_4$)[39] and 50 μg ml$^{-1}$ kanamycin. Cultures were grown at 30 °C with 220 r.p.m. shaking to an optical density at 600 nm (OD$_{600}$) of 0.8 and then cooled to 20 °C for 1 h. To induce expression, isopropyl-β-D-thiogalactopyranoside (IPTG) was added to a final concentration of 100 μg ml$^{-1}$. After 18 h, cells were harvested by centrifugation at 5,000$g$ for 20 min. Cell pellets were flash-frozen in liquid nitrogen before storage at −80 °C.

### XlHAS1 purification

All preparation steps were carried out at 4 °C unless stated otherwise. To purify XlHAS1, typically a 0.9-L culture of cell suspension was thawed and diluted to 200 ml using buffer A supplemented with 5 mM β-mercaptoethanol (BME), 1 mM phenylmethylsulphonyl fluoride (PMSF), 10 mM imidazole, 1% β-dodecyl maltoside (DDM) and 0.2% cholesteryl hemisuccinate (CHS). Cells were lysed using a tissue homogenizer and rocked for 1 h at 4 °C, followed by ultracentrifugation at 200,000$g$ for 45 min. The cleared lysate was mixed with 10 ml of 50% Ni-NTA (Thermo Fisher) resin suspension equilibrated in buffer A and subjected to batch binding for 1 h. After that, the slurry was poured into a glass gravity flow column (Kimble), the flow-through was discarded and the resin was washed three times with 50 ml of Buffer A supplemented with 0.03% GDN (wash1) and 1 M NaCl (wash2) or 20 mM imidazole (wash3). XlHAS1 was eluted using elution buffer (EB) consisting of 25 mM Tris-HCl pH 7.8, 150 mM NaCl, 10% glycerol, 350 mM imidazole and 0.03% GDN in two steps. First, 15 ml of EB was added followed by ~5-min incubation, draining and addition of another 15 ml of EB and draining. The eluted sample was concentrated using a 50-kDa molecular weight cutoff (MWCO) Amicon centrifugal concentrator (Millipore) to <1 ml for SEC using a Superdex200 column (GE healthcare) equilibrated in gel-filtration buffer 1 (XlHAS1-GFB1) consisting of 20 mM Tris-HCl pH 7.8, 150 mM NaCl and 0.02% GDN. The target peak fractions were pooled and concentrated as necessary for subsequent experiments.

### CvHAS purification

All steps were carried out at 4 °C. Harvested C43 cells were resuspended in RB consisting of 20 mM Tris-HCl pH 7.5, 100 mM NaCl, 10% glycerol and 0.5 mM tris(2-carboxyethyl)phosphine (TCEP). The suspension was Dounce-homogenized and mixed with 1 mg ml$^{-1}$ egg white lysozyme for 1 h. Cells were lysed by three passages through a microfluidizer at 18,000 psi; 2 mM PMSF was added after the first passage. Unbroken cells and debris were cleared by centrifugation at 20,000$g$ for 10 min. Supernatant was collected to isolate membranes by ultracentrifugation at 200,000$g$ for 2 h. The crude membrane fraction was Dounce-homogenized in SB (20 mM Tris-HCl pH 7.5, 300 mM NaCl, 40 mM imidazole, 10% glycerol, 0.5 mM TCEP, 1% DDM, 0.1% CHS and 1 mM PMSF) and mixed for 1 h by gentle inversion. Insoluble material was pelleted by ultracentrifugation at 200,000$g$ for 30 min. The supernatant was mixed in batch with 5 ml of Ni-NTA resin (Thermo Fisher), equilibrated in SB, for 1 h. Flow-through material was collected by gravity, after which the nickel resin was washed with 20 bed volumes of WB1 (20 mM Tris-HCl pH 7.5, 1 M NaCl, 10% glycerol, 40 mM imidazole, 0.02% DDM, 0.002% CHS and 0.5 mM TCEP) and 20 volumes of WB2 (20 mM Tris-HCl pH 7.5, 300 mM NaCl, 10% glycerol, 80 mM imidazole, 0.02% DDM, 0.002% CHS and 0.5 mM TCEP). CvHAS was eluted in five volumes of EB (20 mM Tris-HCl pH 7.5, 300 mM NaCl, 10% glycerol, 320 mM imidazole, 0.02% DDM, 0.002% CHS and 0.5 mM TCEP). The eluate was concentrated and injected on an S200 Increase 10/300 GL column (Cytiva) equilibrated in CvHAS-GFB1 (20 mM Tris-HCl pH 7.5, 100 mM NaCl, 0.5 mM TCEP, 0.02% DDM and 0.002% CHS). Fractions containing CvHAS were pooled and concentrated for subsequent reconstitution.

### Nanobody expression and purification

CvHAS nanobodies were expressed and purified as described previously[13]. Briefly, pMESy4 vectors carrying genes for Nb872, Nb881 and Nb886 were transformed into *E. coli* WK6 cells. LB broth supplemented with 100 μg ml$^{-1}$ ampicillin, 1 mM MgCl$_2$ and 0.1% D-glucose was inoculated from appropriate glycerol stocks for overnight precultures. Two 1-L TB flasks supplemented with 1xM salts, 100 μg ml$^{-1}$ ampicillin, 0.4% glycerol, 1 mM MgCl$_2$ and 0.1% D-glucose were inoculated with 2 ml of overnight preculture and incubated at 37 °C with 220 r.p.m. shaking until OD$_{600}$ = 0.7. The shaker temperature was reduced to 28 °C following the addition of IPTG to 1 mM final concentration. After 18 h, cultures were harvested and pellets were flash-frozen in liquid nitrogen and stored at −80 °C.

To purify CvHAS nanobodies, 2-L cell pellets were mixed with 25 ml of 1× TES buffer (200 mM Tris-HCl pH 7.5, 0.5 mM EDTA and 500 mM sucrose) for 30 min. The resulting periplasmic extract was diluted threefold with 0.25× TES and mixed for an additional 30 min at 4 °C. The extraction mixture was cleared of undesired cell debris by centrifugation at 200,000$g$ for 30 min. The supernatant was batch mixed with 5 ml of Ni-NTA beads for 1 h. Unbound material was collected as flow-through over a gravity column and the resin was washed with Nb-WB1 (20 mM Tris-HCl pH 7.5, 1 M NaCl and 20 mM imidazole). Nanobodies were eluted with 300 mM imidazole and injected on an S75 gel-filtration column (Cytiva) equilibrated in Nb-GFB (20 mM Tris-HCl pH 7.5 and 100 mM NaCl). Nanobody-containing fractions were flash-frozen and stored at −80 °C.

### CvHAS inverted membrane vesicle (IMV) preparation

To prepare CvHAS-bearing IMVs, a 2-L cell pellet was resuspended in buffer A (20 mM Tris-HCl pH 7.5, 100 mM NaCl, 10% glycerol, 5 mM BME and 2 mM PMSF). Cells were incubated with 1 mg ml$^{-1}$ lysozyme for 1 h before disruption by three passes through a microfluidizer at 18,000 psi. Unbroken cells were pelleted by centrifugation at 20,000$g$ for 25 min. Cleared lysate was carefully layered onto 40 ml of 2 M sucrose, followed by centrifugation at 200,000$g$ for 2 h. The brown IMV ring was collected, diluted to 60 ml using buffer B (20 mM Tris-HCl

pH 7.5, 100 mM NaCl and 10% glycerol) and subjected to another round of ultracentrifugation for 1 h. The pelleted IMVs were resuspended in 2 ml of Buffer B, aliquoted, flash-frozen and stored at −80 °C for subsequent use in activity assays.

## Quantification of HAS activity by scintillation counting

All reagents for biochemical analyses were supplied by Sigma, unless stated otherwise. Protein concentrations in CvHAS IMVs were normalized as described previously[40]. Reaction buffer consisted of 40 mM Tris-HCl pH 7.5, 150 mM NaCl, 20 mM MnCl$_2$, 0.5 mM TCEP, 5 mM UDP-GlcNAc, 5 mM UDP-GlcA and 0.01 µCi µl$^{-1}$ [$^3$H]UDP-GlcNAc (Perkin Elmer). Reactions were carried out at 30 °C for 2 h. HA digests were performed by adding 2% DDM and 75 U of hyaluronidase (MP Biotech) to the reaction and incubating for an additional 10 min at 30 °C. Reactions were terminated with 3% SDS and product accumulation was quantified using descending paper chromatography and liquid scintillation counting as previously described[40].

## Electrophoretic HA size determination

To assess the size of HA synthesized by XlHAS1, similar reaction conditions were applied to those described above. Reaction buffer consisted of 25 mM Tris-HCl pH 7.8, 150 mM NaCl, 0.02% GDN, 20 mM MgCl$_2$, 5 mM UDP-GlcA and 5 mM UDP-GlcNAc. Reactions were carried out at 37 °C for 3 h and at 1 µM HAS concentration. Controls were carried out by digesting the synthesized polymers with hyaluronidase. Synthesis reactions were mixed with SDS–PAGE loading dye and applied to a 1% agarose gel (ultrapure agarose, Invitrogen) casted in an Owl B2 system (Thermo Fisher). All gels were subjected to electrophoresis at 100 V for 2 h at room temperature to achieve comparable separation for each run. After the run, the gel was equilibrated in 50% ethanol for 1.5 h and subjected to staining in 0.05% Stains-all (Sigma) in 50% ethanol overnight under light protection. The poststaining background was reduced by soaking the gel in 20% ethanol for 3–7 days in the dark. The migration of the synthesized HA species was compared to HA standards from *Streptococcus equi* (Sigma), as well as enzymatically generated ladders: HA LoLadder and HA HiLadder (Hyalose).

## HA extraction from *Chlorella* algae

All reagents used for HA extraction were acquired from Sigma-Aldrich unless stated otherwise. First, 8 ml of PBCV-1-infected NC64A *Chlorella* algae at 7 × 10$^8$ cells per mL were mixed 1:1 with extraction buffer (2 mM Tris-HCl pH 7.5, 5 mM EDTA, 20 mM NaCl, 0.1% SDS, 0.2 mg ml$^{-1}$ DNase, 0.2 mg ml$^{-1}$ RNase and 0.2 mg ml$^{-1}$ Driselase) by rocking for 1 h at room temperature. After 1 h, proteinase K was added to a final concentration of 0.1 mg ml$^{-1}$ and the extract was mixed overnight. Algal extract was mixed with nine volumes of 2:1 CHCl$_3$ and methanol by vigorous shaking. Three volumes of CHCl$_3$ and three volumes of Milli-Q-purified water (MQ-H$_2$O) were added followed by vigorous mixing to complete aqueous extraction. Phase separation was allowed to occur under gravity for 1 h. The aqueous fraction was removed using a serological pipette and mixed with ice-cold isopropanol at a final concentration of 90%. HA precipitation was allowed to occur on ice for 1 h. The HA pellet was isolated by centrifugation at 15,000$g$ for 15 min at 4 °C. The resulting HA pellet was resuspended in 1 ml of MQ-H$_2$O and dried to ~50 µl in a speed-vac to concentrate for agarose gel electrophoresis experiments.

## Substrate turnover rate quantification

UDP release during HA synthesis was quantified using an enzyme-coupled assay as previously described[10,13]. Depletion of reduced nicotinamide adenine dinucleotide (NADH) was monitored at 340 nm every 60 s for 3 h at 37 °C in a SpectraMax instrument using the SoftMax software. The raw data were processed in Microsoft Excel and a linear phase of each reaction was determined. The rate of NADH depletion was converted to µmol of UDP released using a UDP standardized plot (Extended Data Fig. 1c) for subsequent Michaelis–Menten constant

determination using GraphPad Prism. For substrate turnover rate quantification of XlHAS1 mutants, µmol of released UDP was converted to the number of corresponding substrate turnovers per molecule of XlHAS1 per minute. Data are presented relative to WT activity. All experiments were performed in four to eight replicates and error bars represent the s.d. from the mean.

## XlHAS1 reconstitution and identification of specific Fabs

For Fab selection, XlHAS1 was reconstituted into *E. coli* total lipid nanodiscs[41] using chemically biotinylated membrane scaffold protein (MSP) 1D1. The purified enzyme was mixed with MSP and sodium cholate-solubilized lipids at 40 µM final concentration in a 1-ml final volume according to a 1:4:80 molar ratio of HAS, MSP and lipids. Detergent removal was initiated 1 h after mixing all ingredients by adding 200 mg of BioBeads (BioRad) and mixing at 4 °C. After 1 h, another batch of BioBeads was added, followed by mixing overnight. The following morning (after ~12 h), the reconstitution mixture was transferred to a fresh tube and the last batch of BioBeads was added, followed by mixing for 1 h and SEC using Superdex200 column equilibrated with XlHAS1-GFB1 lacking detergent.

Phage selection was performed as previously described[42,43]. In the first round of selection, 400 nM XlHAS1-loaded nanodiscs diluted in selection buffer (20 mM HEPES pH 7.5, 150 mM NaCl and 1% BSA) were immobilized on streptavidin paramagnetic beads (Promega). Beads were washed three times in the selection buffer, with 5 mM D-desthiobiotin added during the first wash to block nonspecific binding. Fab phage library E[44] resuspended in selection buffer was added to the beads and incubated for 1 h with gentle shaking. The beads were washed three times in the selection buffer and then transferred to log-phase *E. coli* XL1-Blue cells. Phages were amplified overnight in 2xYT medium with ampicillin (100 µg ml$^{-1}$) and M13-KO7 helper phage (10$^9$ plaque-forming units per ml). Four additional rounds of selection were performed with decreasing target concentration (200 nM, 100 nM, 50 nM and 25 nM) using a KingFisher magnetic bead handler (Thermo Fisher). In every subsequent round, the amplified phage pool from the previous round was used as the input. Before being used for selection, each phage pool was precleared by incubation with 100 µl of streptavidin magnetic beads. Additionally, 2 µM nonbiotinylated MSP1D1 nanodiscs were present in the selection buffer to reduce the presence of nonspecific binders during rounds 2–5. In these rounds, selection buffer supplemented with 1% Fos-choline-12 was used to release the target and bound phages from the nanodiscs. Cells infected after the last round were plated on LB agar with ampicillin (100 µg ml$^{-1}$) and phagemids from individual clones were sequenced at the University of Chicago Comprehensive Cancer Center Sequencing Facility to identify unique binders. Single-point phage ELISA was used to validate specificity of unique binders as described previously[43]. Fabs were expressed and purified as described[43] and used for activity assays and SEC coelution experiments with XlHAS1. The strongest noninhibitory binders were chosen for cryo-EM trails and one of those yielded well-structured projections of the HAS–Fab complex.

## CvHAS liposome and nanodisc reconstitution

SEC fractions containing CvHAS were pooled for reconstitution. To prepare proteoliposomes, CvHAS was mixed for 30 min at a 3:10,000 ratio with *E. coli* total lipid extract (Avanti) solubilized in DDM. BioBeads (BioRad) were added batchwise to ~1/4 the total reaction volume, with each addition separated by at least 1 h, until the sample became visibly turbid. The mixture was first cleared of protein and lipid aggregates by centrifugation at 21,000$g$ for 10 min. Subsequently, liposomes were isolated by spinning the supernatant at 200,000$g$ for 30 min. Liposome pellets were resuspended in 20 mM Tris-HCl pH 7.5, 100 mM NaCl, 10% glycerol and 0.5 mM TCEP.

For nanodisc preparation, purified CvHAS was combined with MSP1E3D1 and *E. coli* total lipid extract at a 1:3:30 ratio. When

reconstituting CvHAS in a ternary complex with nanobodies, the initial mixture was supplemented with a threefold molar excess of appropriate Nb composition (3:3:1 Nb872, Nb881 and CvHAS or 3:3:1 Nb872, Nb886 and CvHAS). The reconstitution vial was mixed by inversion for 30 min before adding BioBeads to about one fourth of the reaction volume. After 30 min, an additional volume of BioBeads was added and the reconstitution mixture was allowed to incubate overnight. The next day, another volume of BioBeads was added. After 30 min, the mixture was cleared of BioBeads, as well as lipid and protein aggregates, by passing through a 0.2-µm cellulose acetate spin filter. The filtered mixture was reinjected on an S200 SEC column equilibrated in CvHAS-GFB2 (20 mM Tris-HCl pH 7.5, 100 mM NaCl, 2 mM $MnCl_2$ and 0.5 mM TCEP). CvHAS nanodisc fractions were screened by SDS–PAGE for coeluting complex components.

## XlHAS1 cryo-EM sample preparation

GDN-solubilized XlHAS1 was used for cryo-EM experiments. The purified enzyme was mixed with Fab at a 1:4 molar ratio and incubated overnight at 4 °C, followed by SEC using a Superose 6 column (Cytiva) equilibrated with XlHAS1-GFB2 containing 0.01% GDN.

Initial attempts to prepare cryo-EM grids of the purified XlHAS1–Fab complex in the presence of substrates failed to capture an HAS–HA intermediate because of rapid HA accumulation in the sample hampering grid vitrification. To generate an HA-associated sample, substrates (UDP-GlcA and UDP-GlcNAc, 2.5 mM each; Sigma), 20 mM $MgCl_2$ and recombinant HA lyase (0.01 mg ml$^{-1}$; purified as described previously[45]) were included during the overnight Fab incubation, followed by SEC. After SEC, the sample was concentrated to 8 mg ml$^{-1}$ using a 100-kDa MWCO Amicon ultrafiltration membrane. Attempts to generate a GlcNAc-terminated and HA-bound and UDP-bound sample involved the addition of 2.5 mM UDP-GlcNAc and 2.5 mM $MgCl_2$ after SEC and incubation on ice for 1 h before cryo-EM grid vitrification. Then, 4 µl of XlHAS1 sample was applied onto the C-flat 1.2/1.3 grid, glow-discharged for 45 s in the presence of one drop of amylamine. Grids were blotted for 4 s at a blot force of 4 at 4 °C and 100% humidity and plunge-frozen in liquid ethane using Vitrobot Mark IV (FEI). This sample yielded the UDP-bound and gating-loop-inserted XlHAS1 structure.

## CvHAS cryo-EM sample preparation

For the GlcNAc-primed, UDP-GlcA-bound state of CvHAS, 5 mM GlcNAc was included in the initial nanodisc reconstitution mixture. Purified D302N CvHAS nanodiscs in complex with Nb881 and Nb872 were concentrated to 3.5 mg ml$^{-1}$, then further supplemented with 10 mM $MnCl_2$, 5 mM GlcNAc and 5 mM UDP-GlcA and incubated on ice for 10 min. Quantifoil R1.2/1.3 300 mesh grids were glow-discharged for 45 s with one drop of amylamine. A sample volume of 2.5 µl was applied to each grid at 4 °C and 100% humidity, blotted for 12 s with a blot force of 4 and plunge-frozen in liquid ethane using Vitrobot.

To capture the HA disaccharide-bound state, purified CvHAS nanodiscs in complex with Nb881 and Nb872 at 3.0 mg ml$^{-1}$ protein concentration were supplemented with 5 mM GlcNAc, 2 mM $MnCl_2$ and 1 mM UDP-GlcA. GlcNAc extension by UDP-GlcA was allowed to occur on ice for 45 min before 3 µl of sample was applied to glow-discharged QF R1.2/1.3 grids for plunge-freezing.

The sample for CvHAS associated with an HA disaccharide and UDP was obtained by purifying CvHAS nanodiscs in complex with Nb886 and Nb872. The purified complex was concentrated to 7.0 mg ml$^{-1}$ and diluted twofold with a reaction mixture containing 10 mM UDP-GlcNAc, 10 mM UDP-GlcA and 40 mM $MnCl_2$. After mixing, the sample was incubated at room temperature for 15 min before applying 2.5 µl to glow-discharged QF R1.2/1.3 grids for plunge-freezing.

## Cryo-EM data collection and processing

All cryo-EM datasets were collected using the EPU software on a Titan Krios equipped with a K3/GIF detector (Gatan) at the Molecular EM Core (University of Virginia School of Medicine). Then, 40-frame videos were recorded in counting mode at ×81,000 nominal magnification, with a target defocus of −2.0 to −1.0 µm and a total dose of 50 e$^{-}$ per Å$^2$.

All datasets were processed in cryoSPARC[46]. Raw movies were subjected to patch motion correction and patch contrast transfer function estimation. Particles were automatically selected by template picker and sorted by iterative cycles of two-dimensional (2D) classification and heterogeneous refinement. To separate HASs bound to their respective ligands, three-dimensional (3D) variability and 3D classification approaches were used. The final volumes were refined using nonuniform and local refinements to generate high-resolution maps with 3.0–3.3-Å average resolutions (Extended Data Figs. 2–4 and 7–9).

## Model building

To generate the XlHAS1 model, the AF2 (ref. [47]) prediction (B1WB39) was docked into the EM map using Chimera[48] and the model was iteratively real-space-refined in WinCoot[49] and PHENIX[50].

Across all our models, we were able to model most of XlHAS1 residues with the exception of the flexible loop of the GT domain (residues 172–193), as well as the N-terminal (residues 1–14) and C-terminal (residues 569–588) extensions.

Substrate-bound and HA oligo-bound CvHAS models were generated by docking in the proper ligands in Coot using the previously published CvHAS model (Protein Data Bank (PDB) 7SP6), followed by real-space refinement in PHENIX. For the HA2-bound and UDP-bound structure of CvHAS, coordinates for Nb886 were first generated by AF2. Nb886 coordinates were docked into the cryo-EM density map and merged with the CvHAS–Nb872 coordinates. The merged complex and ligand were iteratively real-space-refined.

All model figures were prepared using PyMOL (Schrödinger) or ChimeraX[51].

## Reporting summary

Further information on research design is available in the Nature Portfolio Reporting Summary linked to this article.

## Data availability

Coordinates and EM maps were deposited to the PDB and EM Data Bank under accession codes 8SMM/EMD-40591, 8SMN/EMD-40594, 8SMP/EMD-40598, 8SND/EMD-40623, 8SNC/EMD-40622 and 8SNE/EMD-40624 for XlHAS1 apo, XlHAS1 HA-bound, XlHAS1 UDP-bound, CvHAS GlcNAc and UDP-GlcA-bound, CvHAS GlcA-extended GlcNAc-bound and CvHAS GlcA-extended GlcNAc and UDP-bound, respectively. Source data are provided with this paper.

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

## Acknowledgements

We thank K. Dryden and M. Purdy of the Molecular EM Center at the University of Virginia for support and are indebted to R. Ho for help with cryo-EM data collection. We thank F. Maloney, J. Kuklewicz and L. Wilson for critical comments on the manuscript. We are also grateful to P. DeAngelis (University of Oklahoma Health Sciences Center) for advice on HA detection and providing HA ladders and J. van Etten (Nebraska Center for Virology) for providing *Chlorella* algae. The project was in part funded by National Institutes of Health grants R35GM144130 (to J.Z.) and R01GM117372 (to A.A.K.). I.G. is a recipient of the Boehringer Ingelheim Fonds Fellowship. J.Z. is an investigator of the Howard Hughes Medical Institute (HHMI).

## Author contributions

J.Z., I.G. and Z.S. designed the experiments. I.G. performed all structural and functional analyses of XlHAS1. Z.S. performed all structural and functional analyses of CvHAS. S.K.E., T.G. and A.A.K. selected Fab antibodies against XlHAS1. All authors evaluated and interpreted the data. I.G., Z.S. and J.Z. wrote the paper and all authors edited it.

## Competing interests

The authors declare no competing interests.

## Additional information

**Extended data** is available for this paper at https://doi.org/10.1038/s41594-024-01389-1.

**Correspondence and requests for materials** should be addressed to Jochen Zimmer.

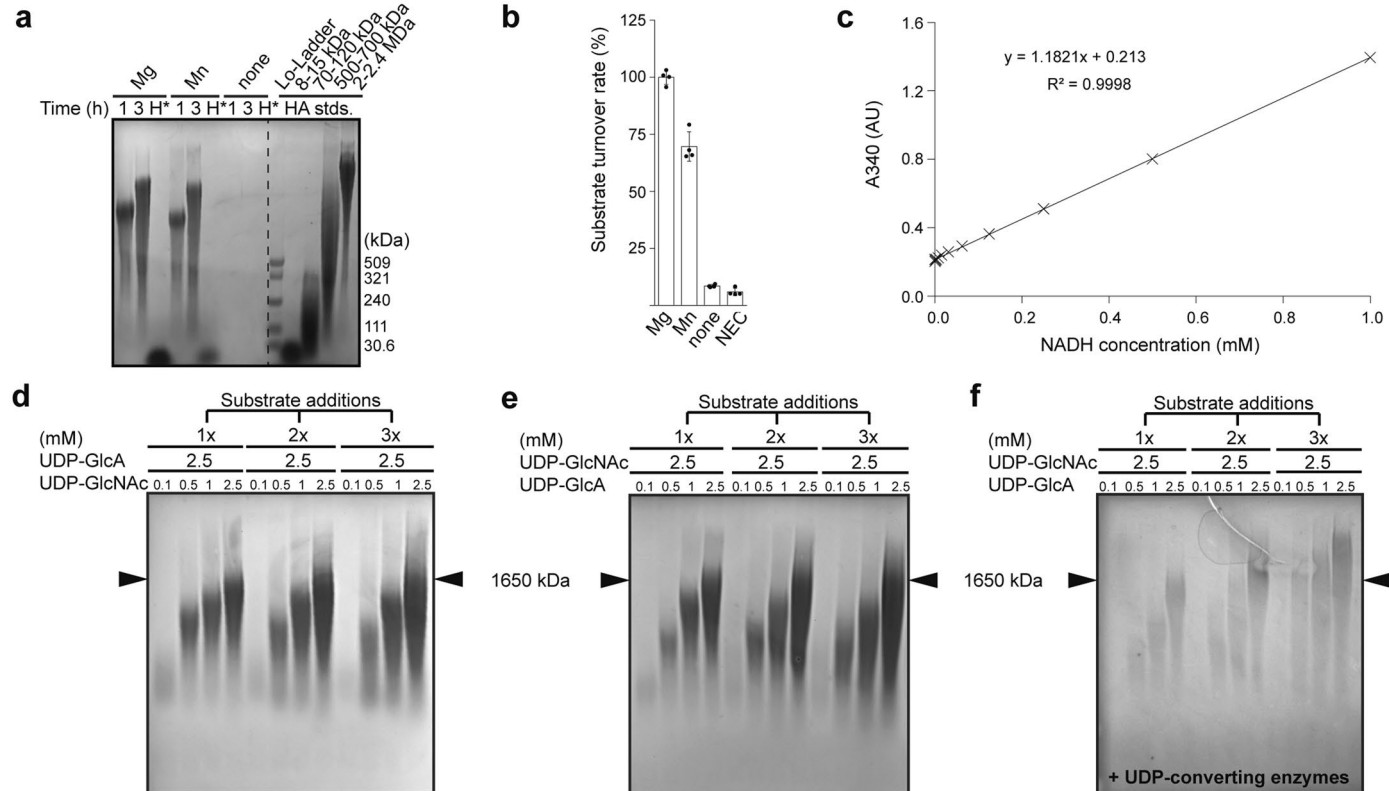

**Extended Data Fig. 1 | *In vitro* catalytic activity of XlHAS1. a,** Agarose gel assay showing HA synthesis is dependent on Mg$^{2+}$, while Mn$^{2+}$ reduces HA production slightly. H* – hyaluronidase digestion. **b,** Same as panel A but by quantifying UDP release. NEC- no enzyme control. This experiment was performed in quadruplicates (n = 4) and error bars represent standard deviations from the means. **c,** NADH calibration curve used for substrate turnover rate calculations. **d,** Substrate replenishing in the absence of UDP-converting enzymes (Lactate dehydrogenase – LDH and pyruvate kinase - PK). UDP-GlcA=2.5 mM and UDP-GlcNAc is varied for this assay. **e,** Same as panel (d), except that UDP-GlcA is varied and UPD-GlcNAc is constant. **f,** Substrate replenishing with LDH/PK present (UDP-GlcNAc=2.5 mM), reverse to what is shown in Fig. 1h. Arrowheads indicate maximum HA extension without UDP removal and at 2.5 mM of both substrates, corresponding roughly to 1.6 MDa HA marker. All experiments were performed at least 3 times with similar results.

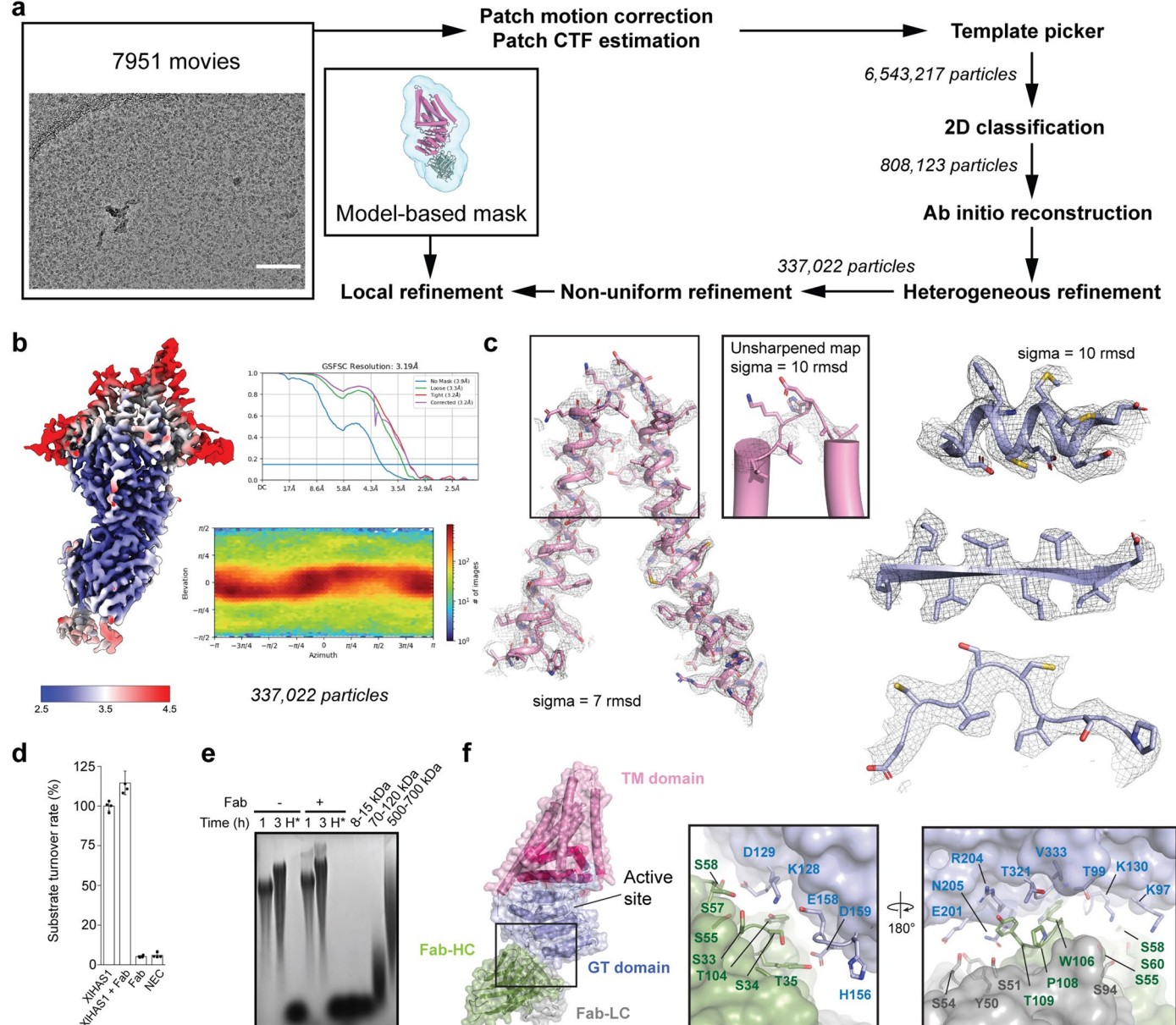

**Extended Data Fig. 2 | CryoEM data processing for apo XlHAS1. a**, CryoSPARC workflow for apo XlHAS1. Scale bar on the micrograph corresponds to 100 nm. **b**, Local resolution of the final map with orientation distribution plot. **c**, Representative regions showing map quality at the indicated contour levels. **d**, Substrate turnover rates in the presence and absence of the Fab. This experiment was performed in quadruplicates (n = 4) and error bars represent standard deviations from the means. **e**, HA synthesis in the presence and absence of the Fab and compared to HA standards of defined molecular weight range. **f**, HAS-Fab interface. Fab binds the GT domain at a position roughly opposite to the active site.

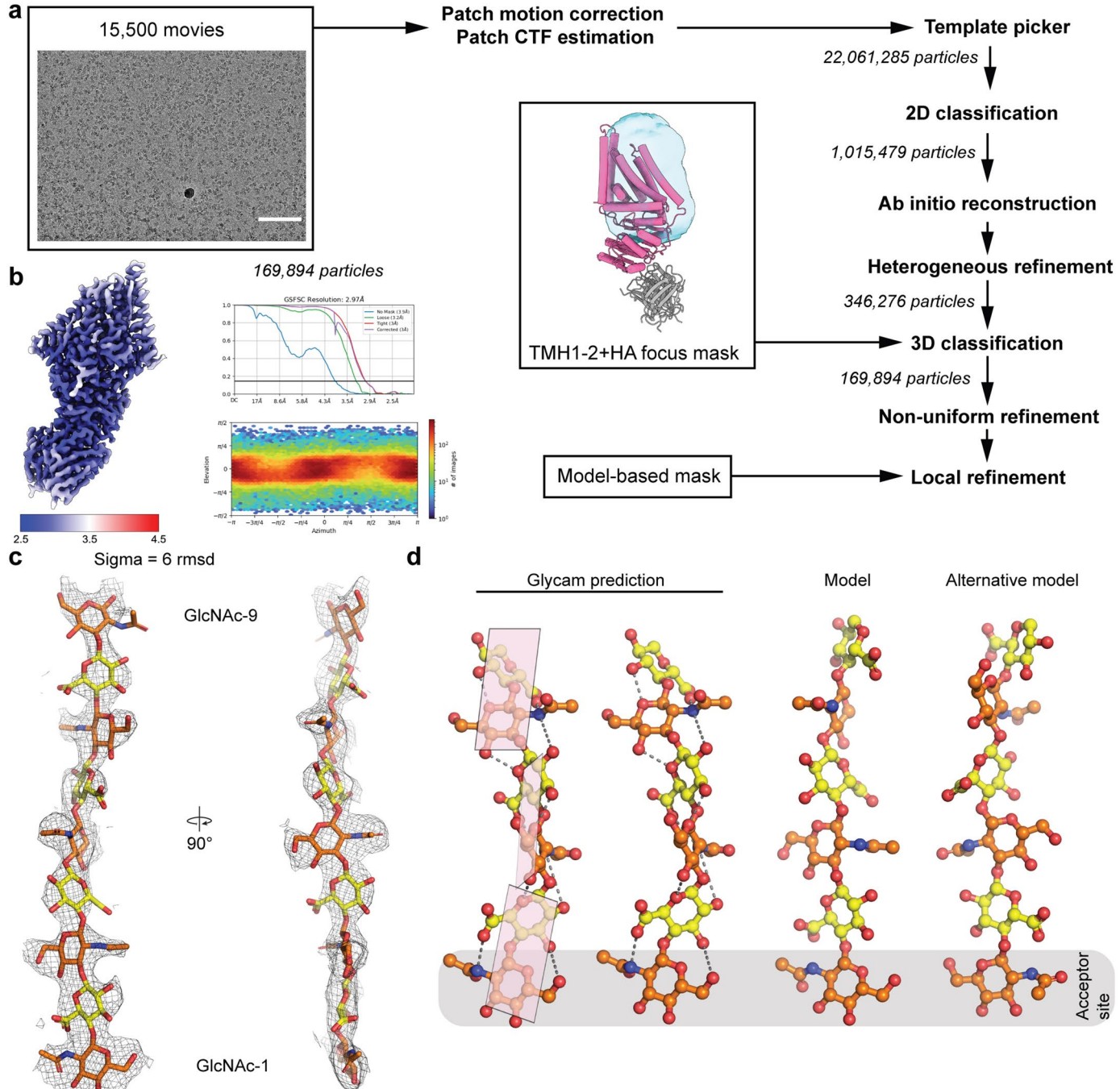

**Extended Data Fig. 3 | CryoEM data processing for HA-bound XlHAS1.**
**a**, CryoSPARC workflow for HA-bound XlHAS1. Scale bar on the micrograph: 100 nm. **b**, Local resolution of the final map with orientation distribution plot. **c**, Map quality for relevant parts of the model. **d**, Different HA conformations.

Left: Glycam predicted structure of an HA hexasaccharide (glycam.org). Middle: Conformation of HA as modeled in the XlHAS1 complex. Right: Alternative conformation of HA with acetamido and carboxylate groups on opposing sides of the polymer.

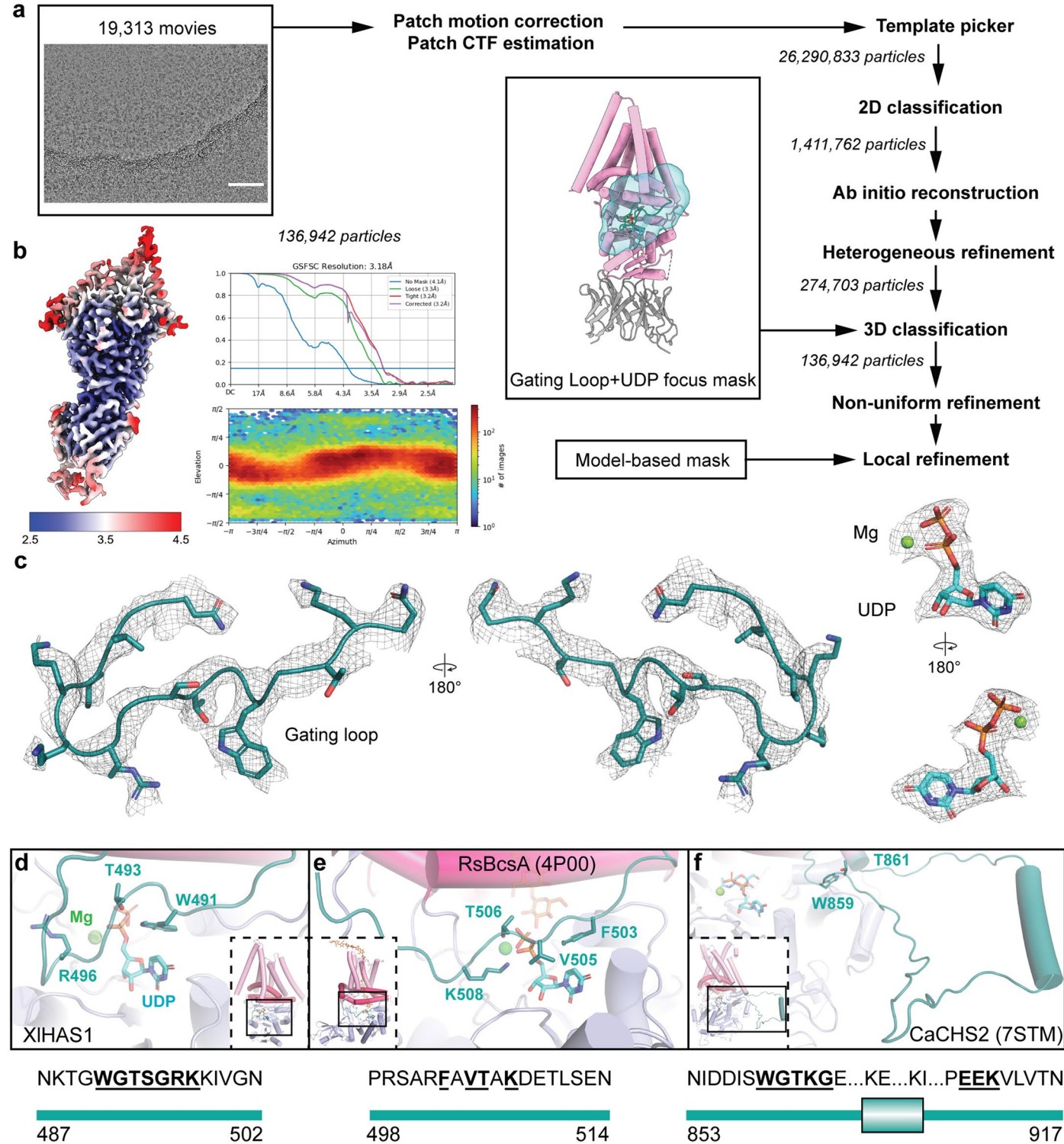

**Extended Data Fig. 4 | CryoEM data processing for UDP-bound XlHAS1.**
**a**, CryoSPARC workflow for UDP-bound XlHAS1. Scale bar on the micrograph corresponds to 100 nm. **b**, Local resolution of the final map with orientation distribution plot. **c**, Map quality for the gating loop and UDP:Mg²⁺. The map was contoured at σ = 7 r.m.s.d. **d**, Position of the gating loop in UDP-bound XlHAS1. **e**, Gating loop in RsBcsA (PDB: 4P00). **f**, Gating loop in *Candida albicans* CHS2 (PDB: 7STL). The gating loop is shown in teal. Numbers indicate gating loop residue ranges. Conserved residues are indicated in bold and underlined font.

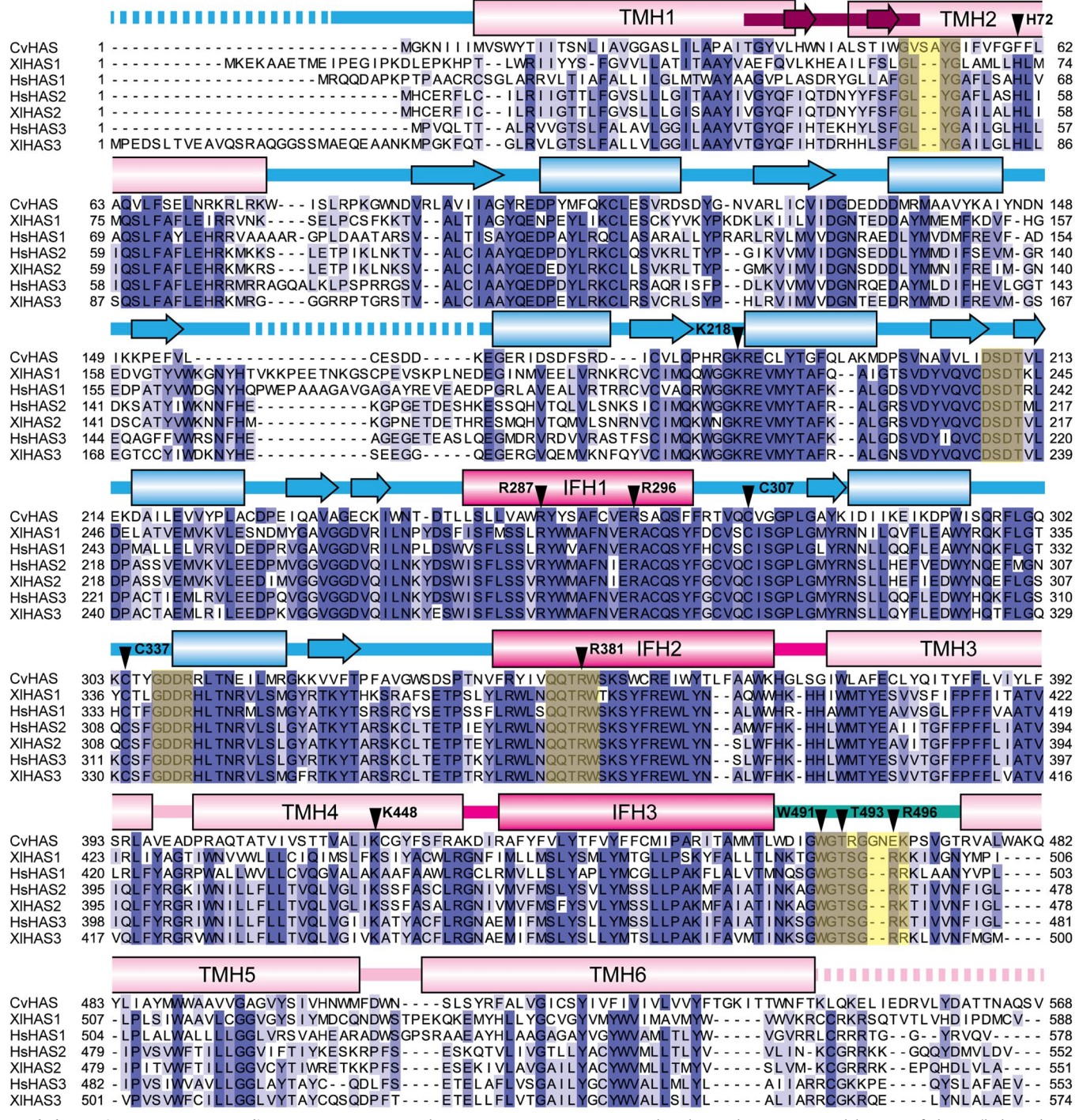

**Extended Data Fig. 5 | HAS sequence alignment.** HAS sequence alignments (MUSCLE)[52] showing conserved motifs (GLYG, DxDT, GDDR, QxxRW and WGTSGRK/R, highlighted in yellow), as well as conserved residues used for mutagenesis studies (indicated with arrowheads). The bar on top of the sequences is colored according to structural domains of XlHAS1 (light pink: TM domain, purple: TMH1-TMH2 loop, pink: IF domain, blue: GT domain, teal: Gating loop). Cylinders indicate α-helices, arrows β-sheets, lines loops, while dashed lines correspond to unstructured regions.

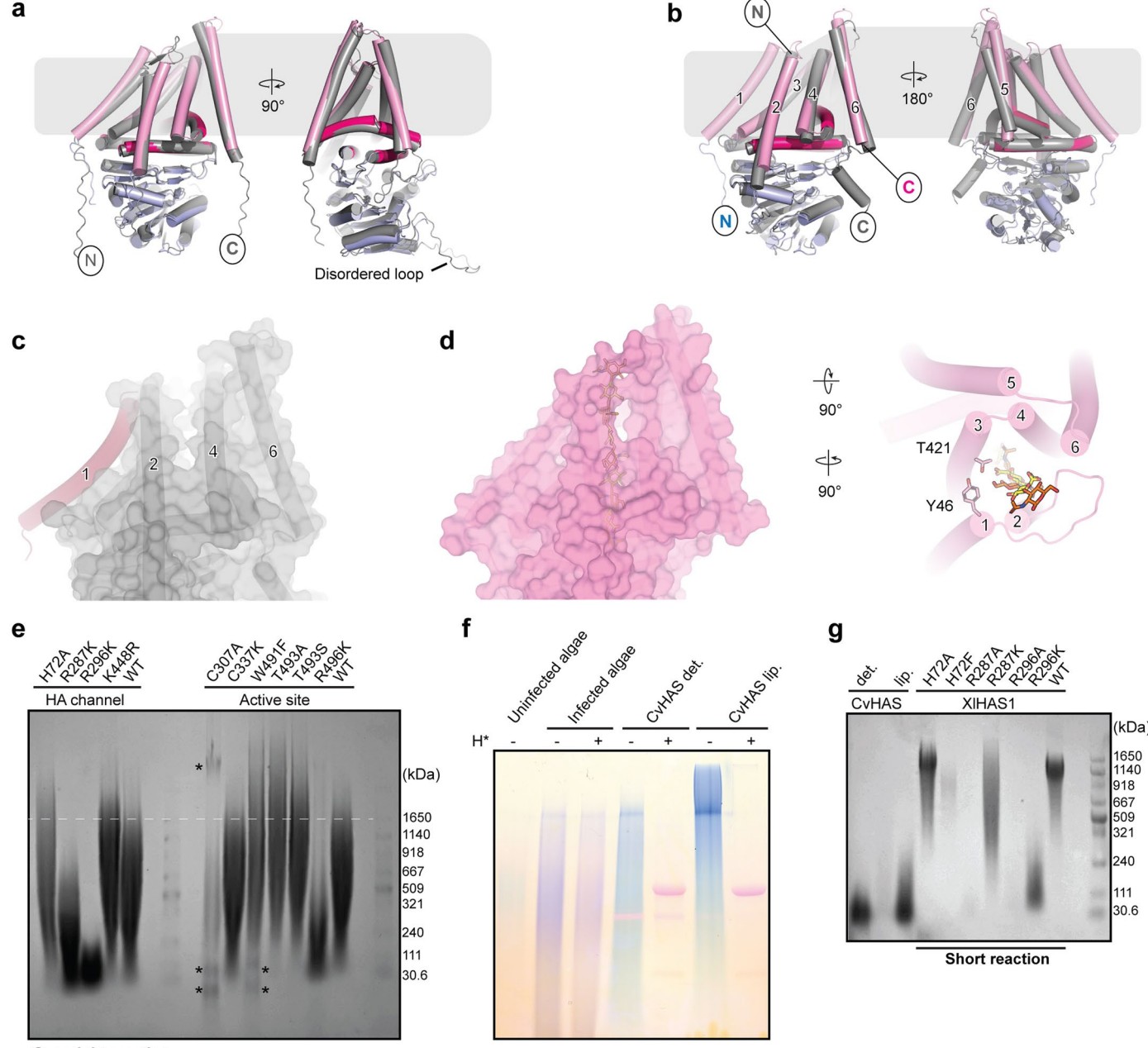

**Extended Data Fig. 6 | HA modeling and gating loop comparison.**
**a**, Superimposition of XlHAS1 apo (colored) and a HsHAS1 AlphaFold2 prediction (gray). N- and C-termini of the polypeptide as well as the HAS1-specific but structurally disordered cytoplasmic loop are indicated. **b**, Superimposition of XlHAS1 apo (colored) and CvHAS apo (gray, PDB: 7SP6). **c**, Surface representation of the CvHAS channel-open state (7SPA) with TMH1 (red) placed according to an AlphaFold 2 prediction. **d**, Surface representation of XlHAS1 in the channel-open state with the nascent HAS polymer completely shielded by the TM region (left panel). Residues stabilizing the TMH1-TMH3 interface (conserved among vertebrate HASs) are shown as sticks (right panel). **e**, HA products for selected XlHAS1 mutants visualized on lower percentage agarose gel (0.5 instead of 1%).

Reactions were carried out overnight. Asterisks indicate protein bands. **f**, Comparison of HA products obtained from NC64A *Chlorella* algae infected with PBCV-1 virus, and *in vitro* synthesized by purified CvHAS in LMNG/CHS micelle (det.) and proteoliposome (lip.). Samples were analyzed on a 4-20% SDS-PAGE gel scanned in full color to show specific hyaluronidase (H*) degradation of the obtained HA (blue smears) in crude algae extracts. DNA and protein contaminations are present (violet and pink smears/bands). SDS-PAGE gels show poor resolution for HA and the ladders were omitted. **g**, Comparison of products synthesized by CvHAS and LMW HA-producing XlHAS1 mutants. Gels shown in panels e-g were stained with Stains-all but only panel (f) is shown in color.

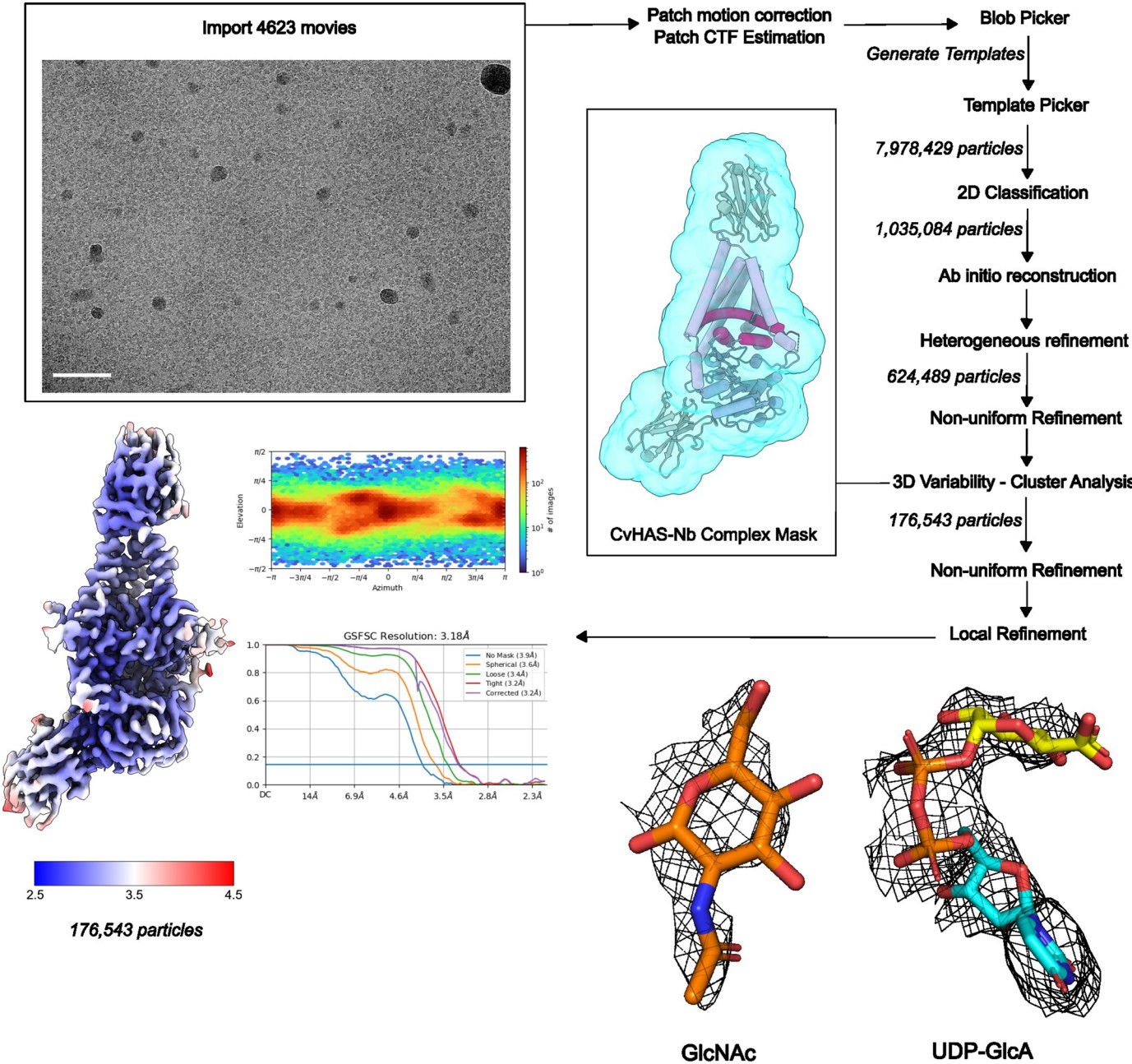

**Extended Data Fig. 7 | CryoEM data processing for UDP-GlcA-bound CvHAS.** CryoEM data processing workflow and map quality for GlcNAc-primed, UDP-GlcA-bound CvHAS. UDP-GlcA and GlcNAc densities are contoured at σ = 4.5 r.m.s.d. Scale bar in electron micrograph = 70 nm.

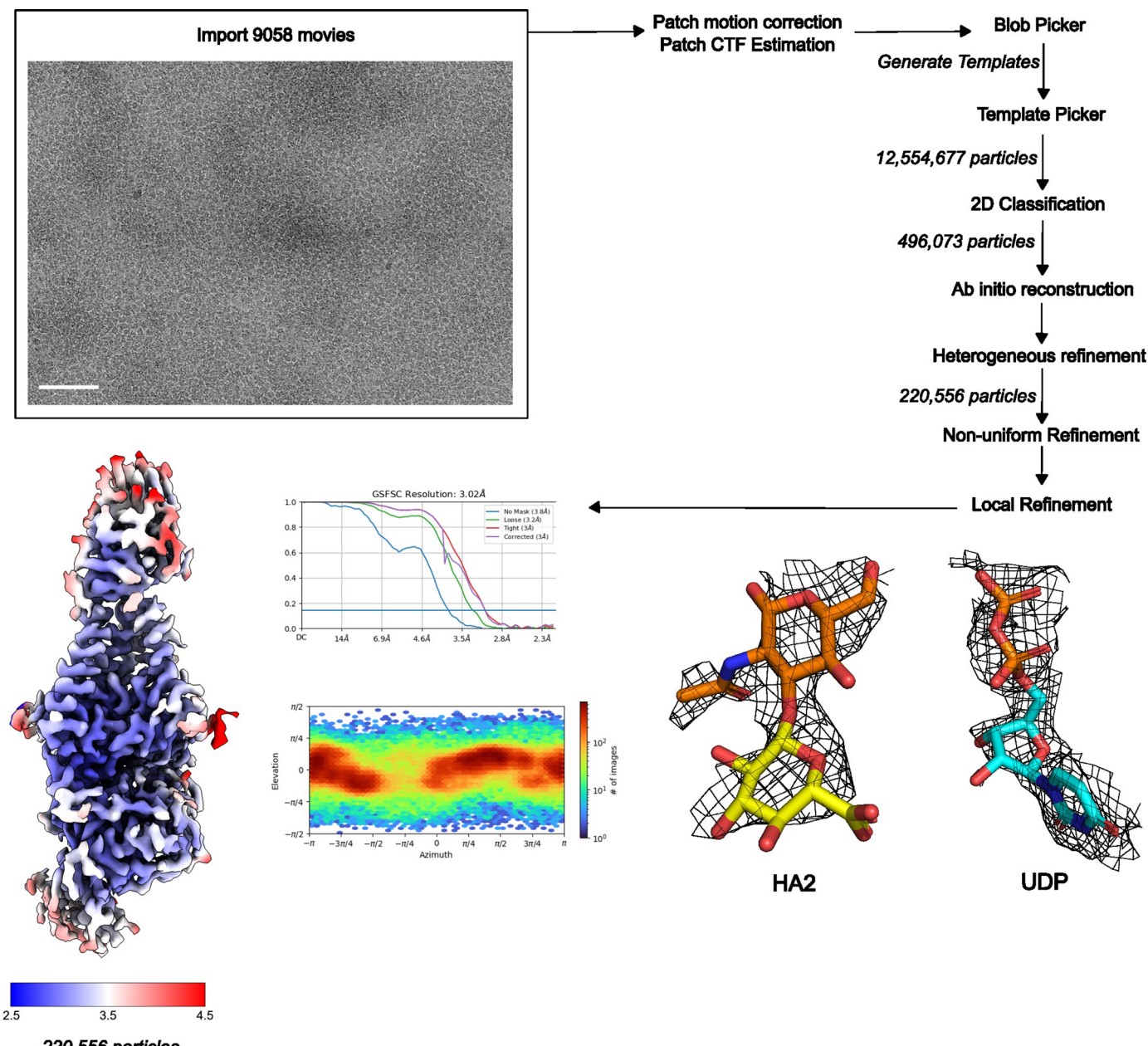

**Extended Data Fig. 8 | CryoEM data processing for disaccharide and UDP-bound CvHAS.** CryoEM data processing workflow and map quality for disaccharide (HA2) and UDP-associated CvHAS. HA2 and UDP densities contoured at σ = 4.5 r.m.s.d. Scale bar in electron micrograph = 70 nm.

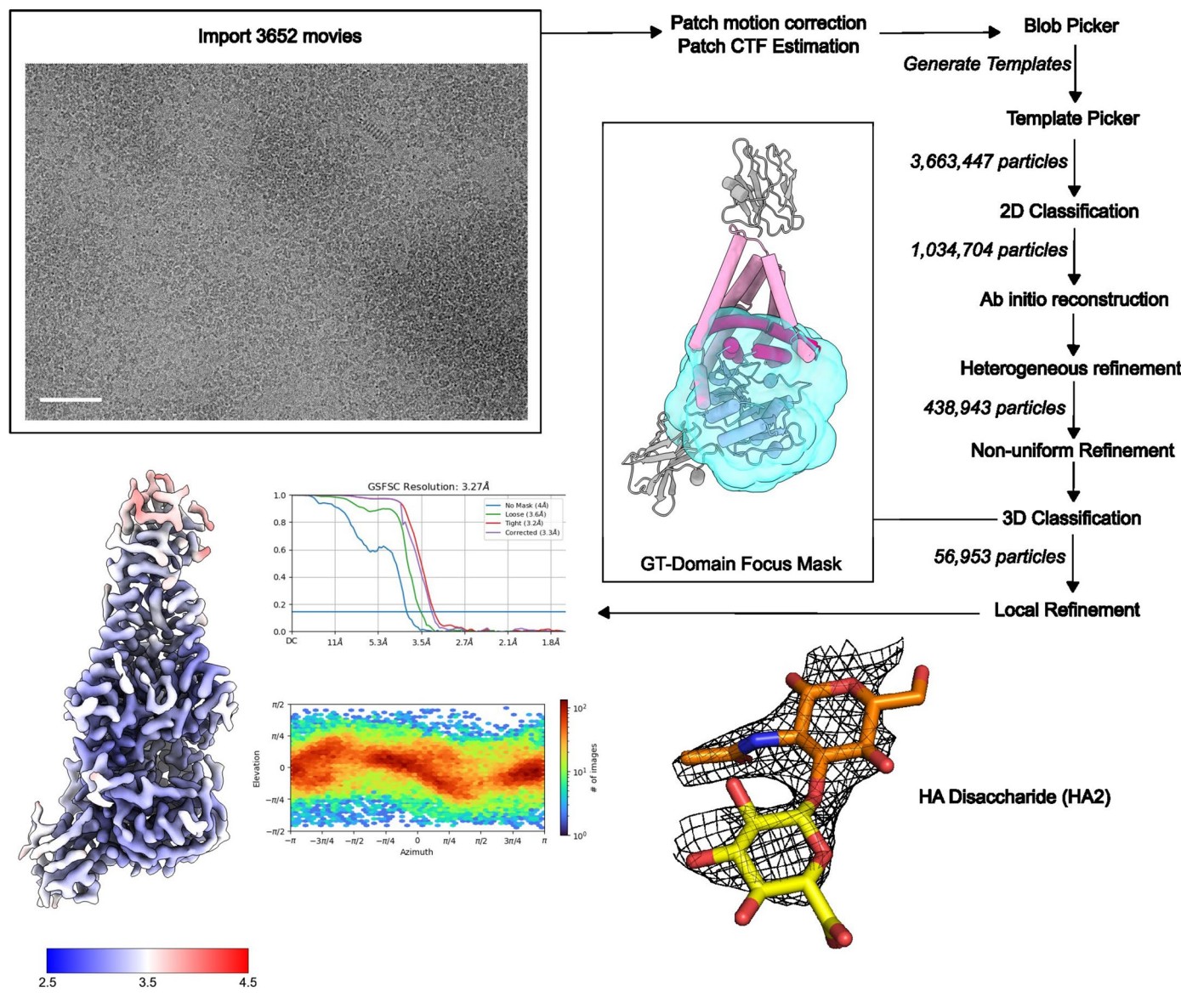

**Extended Data Fig. 9 | CryoEM data processing for disaccharide-bound CvHAS.** CryoEM data processing workflow and map quality for disaccharide (HA2)-associated CvHAS. HA2 density contoured at σ = 4.5 r.m.s.d. Scale bar in electron micrograph = 60 nm.

# Reporting Summary

## Statistics

For all statistical analyses, confirm that the following items are present in the figure legend, table legend, main text, or Methods section.

| n/a | Confirmed | |
|---|---|---|
| ☐ | ☒ | The exact sample size (*n*) for each experimental group/condition, given as a discrete number and unit of measurement |
| ☐ | ☒ | A statement on whether measurements were taken from distinct samples or whether the same sample was measured repeatedly |
| ☒ | ☐ | The statistical test(s) used AND whether they are one- or two-sided<br>*Only common tests should be described solely by name; describe more complex techniques in the Methods section.* |
| ☒ | ☐ | A description of all covariates tested |
| ☒ | ☐ | A description of any assumptions or corrections, such as tests of normality and adjustment for multiple comparisons |
| ☐ | ☒ | A full description of the statistical parameters including central tendency (e.g. means) or other basic estimates (e.g. regression coefficient) AND variation (e.g. standard deviation) or associated estimates of uncertainty (e.g. confidence intervals) |
| ☒ | ☐ | For null hypothesis testing, the test statistic (e.g. *F*, *t*, *r*) with confidence intervals, effect sizes, degrees of freedom and *P* value noted<br>*Give P values as exact values whenever suitable.* |
| ☒ | ☐ | For Bayesian analysis, information on the choice of priors and Markov chain Monte Carlo settings |
| ☒ | ☐ | For hierarchical and complex designs, identification of the appropriate level for tests and full reporting of outcomes |
| ☒ | ☐ | Estimates of effect sizes (e.g. Cohen's *d*, Pearson's *r*), indicating how they were calculated |

*Our web collection on statistics for biologists contains articles on many of the points above.*

## Software and code

Policy information about availability of computer code

| Data collection | EPU 2.5 (ThermoFisher) was used to collect cryoEM data. SoftMax 7.2 (MolecularDevices) software was used to record UDP release curves. |
|---|---|
| Data analysis | CryoEM data was processed in cryoSPARC 4.4 and analyzed in Chimera 1.17.1. Model refinement was performed in Phenix 1.21 and WinCoot 0.9.8.93. ChimeraX 1.6 and PyMOL 2.5.5 were used for cryoEM and model visualization. Catalytic activity data was processed in MS Excel 365 and GraphPad Prism 6. |

For manuscripts utilizing custom algorithms or software that are central to the research but not yet described in published literature, software must be made available to editors and reviewers. We strongly encourage code deposition in a community repository (e.g. GitHub). See the Nature Portfolio guidelines for submitting code & software for further information.

## Data

Policy information about availability of data

All manuscripts must include a data availability statement. This statement should provide the following information, where applicable:
- Accession codes, unique identifiers, or web links for publicly available datasets
- A description of any restrictions on data availability
- For clinical datasets or third party data, please ensure that the statement adheres to our policy

Previously determined structure of CvHAS (PDB 7SP6) was used to build CvHAS models determined in this study. XlHAS1 Alphafold2 prediction (B1WB39) was used to build XlHAS1 apo model. Coordinates and EM maps have been deposited at the Protein Data Bank and Electron Microscopy Data Bank under accession codes

8SMM/EMD-40591, 8SMN/EMD-40594, 8SMP/EMD-40598, 8SND/EMD-40623, 8SNC/EMD-40622, 8SNE/EMD-40624 for Xl-HAS-1 apo, Xl-HAS-1 HA-bound, Xl-HAS-1 UDP-bound, Cv-HAS GlcNAc and UDP-GlcA-bound, Cv-HAS GlcA-extended GlcNAc-bound, Cv-HAS GlcA-extended GlcNAc and UDP-bound, respectively.

# Research involving human participants, their data, or biological material

Policy information about studies with human participants or human data. See also policy information about sex, gender (identity/presentation), and sexual orientation and race, ethnicity and racism.

| | |
|---|---|
| Reporting on sex and gender | n/a |
| Reporting on race, ethnicity, or other socially relevant groupings | n/a |
| Population characteristics | n/a |
| Recruitment | n/a |
| Ethics oversight | n/a |

Note that full information on the approval of the study protocol must also be provided in the manuscript.

# Field-specific reporting

Please select the one below that is the best fit for your research. If you are not sure, read the appropriate sections before making your selection.

☒ Life sciences  ☐ Behavioural & social sciences  ☐ Ecological, evolutionary & environmental sciences

For a reference copy of the document with all sections, see nature.com/documents/nr-reporting-summary-flat.pdf

# Life sciences study design

All studies must disclose on these points even when the disclosure is negative.

| | |
|---|---|
| Sample size | No sample size calculation was performed as the experiments were performed in vitro and the data was reproducible. Each activity assay was performed in at least three technical replicates to calculate standard deviations. |
| Data exclusions | No biochemical data was excluded. Low quality cryoEM datasets were excluded. |
| Replication | Each activity assay was performed in biological triplicates or quadruplicates and each experiment was repeated at least three times. Multiple cryoEM datasets on multiple samples were collected for the individual HAS states. Datasets resulting in the highest quality maps are reported in this study. |
| Randomization | Not applicable as this is not a clinical study. |
| Blinding | Not applicable as this is not a clinical study. |

# Reporting for specific materials, systems and methods

We require information from authors about some types of materials, experimental systems and methods used in many studies. Here, indicate whether each material, system or method listed is relevant to your study. If you are not sure if a list item applies to your research, read the appropriate section before selecting a response.

## Materials & experimental systems

| n/a | Involved in the study |
|---|---|
| ☐ | ☒ Antibodies |
| ☐ | ☒ Eukaryotic cell lines |
| ☒ | ☐ Palaeontology and archaeology |
| ☒ | ☐ Animals and other organisms |
| ☒ | ☐ Clinical data |
| ☒ | ☐ Dual use research of concern |
| ☒ | ☐ Plants |

## Methods

| n/a | Involved in the study |
|---|---|
| ☒ | ☐ ChIP-seq |
| ☒ | ☐ Flow cytometry |
| ☒ | ☐ MRI-based neuroimaging |

## Antibodies

| | |
|---|---|
| Antibodies used | Commercially available primary anti-His (Qiagen, cat. # 34660, diluted 1:1000) and secondary DyLight800-conjugated anti-Mouse (Thomas Scientific, cat. # 610-145-002-0.5, diluted 1:10000) antibodies for Western Blots. |
| Validation | From manufacturer, see https://www.qiagen.com/us/products/discovery-and-translational-research/protein-purification/taggedprotein-expression-purification-detection/anti-his-antibodies-bsa-free?catno=34660 |

## Eukaryotic cell lines

Policy information about cell lines and Sex and Gender in Research

| | |
|---|---|
| Cell line source(s) | Spodoptera frugiperda (Sf9) insect cells were purchased from Expression Systems. |
| Authentication | Non authenticated. |
| Mycoplasma contamination | Not tested for mycoplasma contamination. |
| Commonly misidentified lines (See ICLAC register) | None used. |

## Plants

| | |
|---|---|
| Seed stocks | n/a |
| Novel plant genotypes | n/a |
| Authentication | n/a |

