## [Peer Review File · Nature Structural & Molecular Biology]

Peer Review Information

Manuscript Title: Structural insights into translocation and tailored synthesis of hyaluronan

Corresponding author name(s): Jochen Zimmer

Editorial Notes:

Transferred manuscripts This manuscript has been previously reviewed at another journal that is not operating a transparent peer review scheme. This document only contains reviewer comments, rebuttal and decision letters for versions considered at Nature Structural & Molecular Biology.

Reviewer Comments & Decisions:

Decision Letter, initial version:

Message: Our ref: NSMB-A49062-T

23rd Apr 2024

Dear Dr. Zimmer,

Thank you for submitting your revised manuscript "Structural insights into translocation and tailored synthesis of hyaluronan" (NSMB-A49062-T). It has now been seen by the original referees and their comments are below. The reviewers find that the paper has improved in revision, and therefore we'll be happy in principle to publish it in Nature Structural & Molecular Biology, pending minor revisions to satisfy the referees' final requests and to comply with our editorial and formatting guidelines.

To facilitate our work at this stage, it is important that we have a copy of the main text as

a word file. If you could please send along a word version of this file as soon as possible, we would greatly appreciate it; please make sure to copy the NSMB account (cc'ed above).

Sincerely,

Katarzyna Ciazynska, PhD
(she/her)
Associate Editor
Nature Structural & Molecular Biology
<https://orcid.org/0000-0002-9899-2428>

Reviewer #2 (Remarks to the Author):

Compared to the previous version, the authors have made minor changes to the manuscript, especially in the discussion section. I believe that the necessary changes are in place. I have the following suggestions, which are mostly minor, but need to be addressed before publication.

1. The manuscript has toned down from the "length control mechanism" to "molecular basis for HAS engineering" and changed the title to "tailored synthesis". I think this is ok.
2. The comprehensive comparison between xIHAS and CvHAS. The figures have been modified in Extended data Fig 6. The differences have been summarized as "Comparing the N-terminal regions of XIHAS1 and CvHAS1 shows that (a) TM1 is missing in CvHAS, (b) TM2 is not as 'open' (i.e. bent outwards) in CvHAS as it is in XIHAS1, and (c) the loop connecting TM1 and TM2 is also disordered in CvHAS, instead of closing a lateral opening towards the bilayer, as observed in XIHAS1. Some loop residues are within 3-5 Å from the translocating HA polymer." I think it may be better if the authors can incorporate these before they draw the conclusion such as "CvHAS' poor processivity is founded in differences within its transmembrane architecture, evidenced by structural flexibility of its N-terminal region. We refer to this as differences in architectural dynamics."
3. Still regarding point 2 in the rebuttal letter, the question "However, it remains unclear how HA coordination differs across the three tested conditions: detergent micelle, IMV, and algae in vivo." is not answered by the authors. The authors need to explain what the expected effect of these different biochemical condition on the dynamics of the TM region is and how they correlate to the experimental observation of the product length.
4. The modified Figure 6 seems good to me. However, the old figure 6 which has been moved into supplementary discussion still needs some modification. The comment "the blue and white sugar should be labeled as GlcNAc and GlcA, respectively, instead of being labeled as blue and blue/white squares." in the rebuttal letter is not addressed. The labels for GlcNAc and GlcA have not been revised. Please see the figure below and just label the blue sugar GlcNAc, the white sugar GlcA and remove the squares.

5. The method for liposome reconstitution is missing. Previously, the reviewer thought that the CvHAS lip in Extended data figure 6f corresponds to the IMV preparation in the method session but based on the rebuttal letter, this is different from IMV. Therefore, the authors need to provide the method accordingly.

Reviewer #3 (Remarks to the Author):

Gorniak et al analyze the mechanism of hyaluronate synthase using cryo electron microscopy, and a variety of assays.

The authors have addressed all of my remaining concerns. Framing the issue in terms of elongation rather than length control bypasses any issues around substrate release. And I agree that their mechanistic suggestion, while complex and currently unproven, is at least plausible and testable. Ultimately this paper represents a substantial amount of effort and moves the field forwards considerably.

Author Rebuttal to Initial comments

Please see below for our response to the reviewers' comments.

Reviewer #2:

Remarks to the Author:

Compared to the previous version, the authors have made minor changes to the manuscript, especially in the discussion session. I believe that the necessary changes are in place. I have the following suggestions, which are mostly minor, but need to be addressed before publication.

1. The manuscript has toned down from the “length control mechanism” to “molecular basis for HAS engineering” and changed the title to “tailored synthesis”. I think this is ok.

> Good

2. The comprehensive comparison between xIHAS and CvHAS. The figures have been modified in Extended data Fig 6. The differences have been summarized as “Comparing the N-terminal regions of XIHAS1 and CvHAS1 shows that (a) TM1 is missing in CvHAS, (b) TM2 is not as ‘open’ (i.e. bent outwards) in CvHAS as it is in XIHAS1, and (c) the loop connecting TM1 and TM2 is also disordered in CvHAS, instead of closing a lateral opening towards the bilayer, as observed in XIHAS1. Some loop residues are within 3-5 Å from the translocating HA polymer.” I think it may be better if the authors can incorporate these before they draw the conclusion such

as “CvHAS’ poor processivity is founded in differences within its transmembrane architecture, evidenced by structural flexibility of its N-terminal region. We refer to this as differences in architectural dynamics.”

> The referenced text is from our rebuttal letter. The same observations are already listed in the Discussion, lines 334-343, thereby explaining our interpretation. It is unclear what the reviewer refers to by “I think it may be better if the authors can incorporate these before they draw the conclusion....”.

3. Still regarding point 2 in the rebuttal letter, the question “However, it remains unclear how HA coordination differs across the three tested conditions: detergent micelle, IMV, and algae in vivo.” is not answered by the authors. The authors need to explain what the expected effect of these different biochemical condition on the dynamics of the TM region is and how they correlate to the experimental observation of the product length.

> The text has been revised, line 251: “This effect is likely caused by various membrane mimetics having stabilizing or destabilizing effect on CvHAS’ transmembrane domain and consequently HA coordination.”

4. The modified Figure 6 seems good to me. However, the old figure 6 which has been moved into supplementary discussion still needs some modification. The comment “the blue and white sugar should be labeled as GlcNAc and GlcA, respectively, instead of being labeled as blue and blue/white squares.” in the rebuttal letter is not addressed. The labels for GlcNAc and GlcA have not been revised. Please see the figure below and just label the blue sugar GlcNAc, the white sugar GlcA and remove the squares.

> The supplemental figure has been revised, as requested. The internationally accepted symbols for GlcNAc and GlcA have been removed.

5. The method for liposome reconstitution is missing. Previously, the reviewer thought that the CvHAS lip in Extended data figure 6f corresponds to the IMV preparation in the method session but based on the rebuttal letter, this is different from IMV. Therefore, the authors need to provide the method accordingly.

> Added

Reviewer #3:

Remarks to the Author:

Gorniak et al analyze the mechanism of hyaluronate synthase using cryo electron microscopy, and a variety of assays.

The authors have addressed all of my remaining concerns. Framing the issue in terms of elongation rather than length control bypasses any issues around substrate release. And I agree that their mechanistic suggestion, while complex and currently unproven, is at least plausible and testable. Ultimately this paper represents a substantial amount of effort and moves the field forwards considerably.

> Thanks

Final Decision Letter:

Message: 14th Aug 2024

Dear Dr. Zimmer,

We are now happy to accept your revised paper "Structural insights into translocation and tailored synthesis of hyaluronan" for publication as an Article in Nature Structural & Molecular Biology.

Due to the importance of these deadlines, we ask that you please let us know now whether

you will be difficult to contact over the next month. If this is the case, we ask you provide us with the contact information (email, phone and fax) of someone who will be able to check the proofs on your behalf, and who will be available to address any last-minute problems.

Your paper will be published online soon after we receive proof corrections and will appear in print in the next available issue. You can find out your date of online publication by contacting the production team shortly after sending your proof corrections.

If you have not already done so, we strongly recommend that you upload the step-by-step protocols used in this manuscript to the Protocol Exchange. Protocol Exchange is an open online resource that allows researchers to share their detailed experimental know-how. All uploaded protocols are made freely available, assigned DOIs for ease of citation and fully searchable through nature.com. Protocols can be linked to any publications in which they are

used and will be linked to from your article. You can also establish a dedicated page to collect all your lab Protocols. By uploading your Protocols to Protocol Exchange, you are enabling researchers to more readily reproduce or adapt the methodology you use, as well as increasing the visibility of your protocols and papers. Upload your Protocols at www.nature.com/protocolexchange/. Further information can be found at www.nature.com/protocolexchange/about.

Please note that *Nature Structural & Molecular Biology* is a Transformative Journal (TJ). Authors may publish their research with us through the traditional subscription access route or make their paper immediately open access through payment of an article-processing charge (APC). Authors will not be required to make a final decision about access to their article until it has been accepted. Find out more about Transformative Journals

Sincerely,

Katarzyna Ciazynska, PhD
(she/her)
Associate Editor
Nature Structural & Molecular Biology
<https://orcid.org/0000-0002-9899-2428>